# Data Efficient Adaptation in Large Language Models via Continuous Low-Rank Fine-Tuning

**Xiao Han**[*]
Zhejiang University of Technology,
Zhejiang Key Laboratory of
Visual Information Intelligent Processing
Hangzhou, China
`hahahenha@gmail.com`

**Zimo Zhao**[*]
City University of Hong Kong
Hong Kong, China
`zmzhao6-c@my.cityu.edu.hk`

**Wanyu Wang**[*]
City University of Hong Kong
Hong Kong, China
`wanyuwang4-c@my.cityu.edu.hk`

**Maolin Wang**
City University of Hong Kong
Hong Kong, China
`Morin.wang@my.cityu.edu.hk`

**Zitao Liu**
Jinan University
Jinan, China
`liuzitao@jnu.edu.cn`

**Yi Chang**
Jilin University
Jilin, China
`yichang@jlu.edu.cn`

**Xiangyu Zhao**[†]
City University of Hong Kong
Hong Kong, China
`xianzhao@cityu.edu.hk`

## Abstract

Recent advancements in Large Language Models (LLMs) have emphasized the critical role of fine-tuning (FT) techniques in adapting LLMs to specific tasks, especially when retraining from scratch is computationally infeasible. Fine-tuning enables LLMs to leverage task- or domain-specific data, producing models that more effectively meet the requirements of targeted applications. However, conventional FT approaches often suffer from catastrophic forgetting and suboptimal data efficiency, limiting their real-world applicability. To address these challenges, this paper proposes **DEAL**, a novel framework that integrates Low-Rank Adaptation (LoRA) with a continuous fine-tuning strategy. By incorporating knowledge retention and adaptive parameter update modules, the framework mitigates the limitations of existing FT methods while maintaining efficiency. Experiments on **15** diverse datasets show that **DEAL** consistently outperforms baseline methods, yielding substantial gains in task accuracy and resource efficiency. These findings demonstrate the potential of our approach to advance continual adaptation in LLMs by enhancing task performance while improving resource efficiency. The source code is publicly available at https://github.com/Applied-Machine-Learning-Lab/DEAL.

## 1 Introduction

The advent of Large Language Models (LLMs) has catalyzed transformative advances in Natural Language Processing (NLP), enabling breakthroughs across healthcare, education, web technologies, and other domains [1–5]. However, training and utilizing these models to evolve to real-world

---

[*]Equal contribution.
[†]Corresponding author.

39th Conference on Neural Information Processing Systems (NeurIPS 2025).

demands remains a critical challenge. Direct fine-tuning of billion-scale-parameter models incurs prohibitive computational costs, creating accessibility barriers for resource-constrained researchers and institutions. Even for small and medium-sized enterprises, it is also difficult to independently implement pre-training of these models. In such scenarios, Parameter-Efficient Fine-Tuning (PEFT) methods—particularly Low-Rank Adaptation (LoRA)—have emerged as practical solutions that leverage pre-trained models by "standing on the shoulders of giants" [6, 7]. By applying low-rank matrix decompositions, LoRA reduces the number of trainable parameters by over 90% while maintaining baseline performance. By selectively updating task-specific subspaces, LoRA enables targeted knowledge integration in non-stationary environments—a capability aligned with the core objectives of continual learning.

As LLMs require ongoing updates to ensure their knowledge remains current and relevant over time, continual learning of LoRA is required to integrate new information while preserving existing capabilities [8]. This approach mitigates catastrophic forgetting by freezing most parameters and restricting updates to a low-rank matrix, allowing LoRA to differentially activate specific knowledge within the model. For instance, a LoRA-based model trained on Wiki-QA [9] can be further refined on TruthfulQA [10] to enhance performance. However, while continual learning of the LoRA module only unlocks up-to-date task-specific outcomes, it may compromise cross-domain performance [11], particularly when smaller, specialized datasets lack the breadth of the original pre-training data [12]. Therefore, could we design a high-level fine-tuning method that maintains excellent performance across all tasks while allowing continuous fine-tuning with small-scale datasets?

Several existing studies have explored this problem. In particular, continuous learning can be achieved through two main strategies: (1) direct model editing, and (2) introducing additional adapters. On the one hand, studies [13–18] have shown that key-value-like structures in the Transformer layers can be directly edited. For example, ROME [13] and MEMIT [14] directly update the key-value-like structures in Transformer layers via causal weight interventions. However, these approaches require massive additional experiments to pinpoint which neurons to edit, making it both inefficient and costly. On the other hand, for LoRA-based LLMs, the process of locating and modifying parameters in the low-rank matrix remains largely opaque [19]. To reduce the complexity of targeting specific parameters, many studies turn to stack additional adapter modules [20–23]. Yet, these modules inevitably impose extra computational overhead. Consequently, interpretability and efficiency remain the two major challenges in applying continuous learning to LoRA-tuned LLMs.

To address these limitations, we introduce **D**ata-**E**fficient **A**daptation via continuous **L**ow-rank fine-tuning (**DEAL**), a method that facilitates efficient knowledge acquisition while preserving the interpretability of model updates. Specifically, we design a wavelet kernel to adaptively preserve core features of historical knowledge in the filtered low-rank matrix while seamlessly incorporating new information. By focusing on core aspects of historical knowledge, DEAL prevents catastrophic forgetting, thus maintaining model performance across multiple tasks. This innovative approach provides a robust framework for continuous learning, making it an effective solution for dynamic data environments. Our contributions of the paper are summarized as follows:

1. We introduce DEAL, an innovative continual learning framework that efficiently utilizes small amounts of new data for continuous learning, thereby avoiding the need for relearning and significantly conserving computing resources.

2. We leverage a wavelet kernel to preserve historical knowledge and deploy differentiated regularization terms to control the knowledge updating process, improving both transparency and efficiency. Additionally, by simply replacing the original low-rank matrices with their fine-tuned counterparts, DEAL ensures that inference time remains unchanged.

3. Comprehensive experiments on 15 multi-task open-source datasets validate the effectiveness and efficiency of our framework. These experiments demonstrate its ability to maintain high performance across different tasks while efficiently managing computational resources.

## 2 Preliminaries

In this section, we give the definitions of LoRA-based LLM at first and then introduce the continual learning for LoRA fine-tuning. Finally, we state the problem that we solve in this paper.

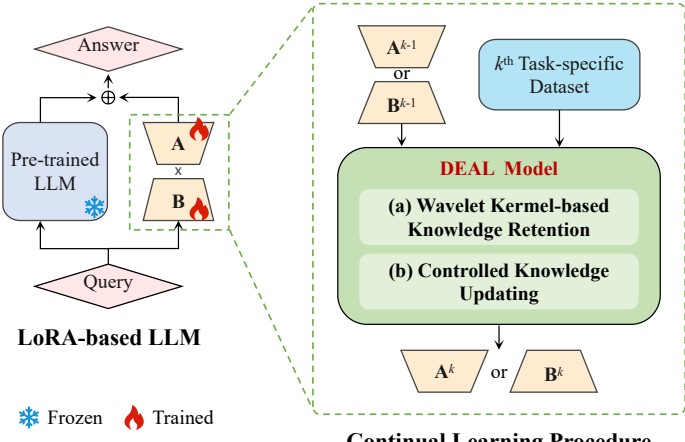

Figure 1: The framework overview.

**LoRA-based LLM.** LoRA enhances LLM by introducing low-rank matrices to weight updates during fine-tuning. Given a pre-trained weight matrix $\boldsymbol{W} \in \mathbb{R}^{m \times n}$, LoRA decomposes the weight update $\Delta \boldsymbol{W}$ into two smaller matrices $\boldsymbol{A} \in \mathbb{R}^{m \times r}$ and $\boldsymbol{B} \in \mathbb{R}^{n \times r}$, where $r$ is a user-defined rank ($r << \min\{m, n\}$). The update is then expressed as $\Delta \boldsymbol{W} = \boldsymbol{A} \times \boldsymbol{B}^{\top}$, allowing the model to adapt to new tasks with significantly fewer parameters compared to full fine-tuning. This decomposition reduces the number of trainable parameters from $m \times n$ to $m \times r + n \times r$, leading to substantial computational savings.

**Continual LoRA Fine-Tuning.** The parameter-efficient fine-tuning approaches of LoRA enable LLMs to learn new tasks while retaining performance on previously learned tasks. By updating only the low-rank matrices $\boldsymbol{A}$ and $\boldsymbol{B}$, the LoRA-based LLM mitigates catastrophic forgetting, a common challenge in continual learning, by enabling the model to learn new tasks while retaining performance on prior tasks.

In this paper, we aim to effectively learn new tasks with acceptable training cost, while retaining performance on previously learned tasks. Mathematically, this involves adjusting the weight matrix $\boldsymbol{W}$ of the model $\mathcal{A}_{\boldsymbol{W}}$ by introducing low-rank updates $\Delta \boldsymbol{W} = \boldsymbol{A} \times \boldsymbol{B}^{\top}$, where $\boldsymbol{A}$ and $\boldsymbol{B}$ are low-rank matrices. The learning target is to minimize the loss function $\mathcal{L}$ over the new task data $\mathcal{D}_{\text{new}}$, subject to a regularization term that penalizes changes to the original parameters to prevent catastrophic forgetting. This can be formulated as:

$$\min_{\boldsymbol{A}, \boldsymbol{B}} \mathcal{L}\left(\mathcal{A}_{\boldsymbol{W} + \boldsymbol{A} \times \boldsymbol{B}^{\top}}, \mathcal{D}_{\text{new}}\right) + \lambda |\boldsymbol{A} \times \boldsymbol{B}^{\top}|, \tag{1}$$

where $\lambda$ is a hyperparameter controlling the trade-off between learning new information and retaining prior knowledge. The goal is to find the optimal low-rank matrices $\boldsymbol{A}$ and $\boldsymbol{B}$ that allow the model to adapt to new tasks while preserving its performance on previous tasks.

## 3 Methodology

In this section, we first provide a framework overview of DEAL and then we introduce each part of DEAL in detail. Finally, we outline the training procedure, which fine-tunes the low-rank matrices to ensure the model adapts to new tasks without sacrificing performance on previously learned tasks.

### 3.1 Framework Overview

Figure 1 shows the overall framework for continuous learning on domain-specific dataset, which consists of a wavelet kernel-based knowledge retention module and a controlled knowledge updating module. **(a) Wavelet Kernel-based Knowledge Retention Module** extracts and filters singular values from the low-rank matrices to preserve the core representations of historical knowledge, which should be maintained throughout continual learning. **(b) Controlled Knowledge Updating Module**

applies higher-order regularization to constrain parameter updates in LoRA, thereby regulating the integration of new knowledge while minimizing disruption to previously learned representations. In LoRA, the small-parameter knowledge representation embedded in the low-rank matrix could activate the corresponding understanding and reasoning abilities in the original LLM, facilitating the learning of new tasks. These two modules allow the model to effectively learn and incorporate new information without disrupting previously acquired knowledge. As for the inference phase, the updated low-rank matrix, with its new knowledge representation, will directly replace the corresponding part in the original LoRA module, ensuring that the inference delay of DEAL remains unaffected.

### 3.2 Wavelet Kernel-based Knowledge Retention

Due to limited hardware resources, continual learning with LoRA-based LLM typically involves fine-tuning only the LoRA module, while the pre-trained LLM parameters remain unchanged. However, updates to the low-rank matrix can increasingly hinder the model's ability to retain the original features crucial for activating the corresponding capabilities within the LLM. To address this, we introduce a wavelet kernel to filter and preserve the key features of LoRA during continual learning.

In the LoRA module, the matrices $A$ and $B$ are singular, meaning they are not full-rank. This presents a challenge for effective feature extraction. We assume that the singular matrix $Y := A$ or $B$ can be decomposed into a task-relevant component $X$ and a redundant or noisy component $D$, i.e., $Y = X + D$.

Here, $X$ denotes the core feature matrix, capturing the intrinsic low-rank structure that encodes task-relevant semantics. According to the Eckart—Young—Mirsky theorem [24], the best low-rank approximation of a matrix in the Frobenius norm sense is achieved via truncation of its singular value decomposition (SVD). This motivates our use of truncated SVD to estimate $X$ from the observed matrix $Y$, with the goal of recovering task-relevant features from its singular representation.

We further assume that both $Y$ and $X$ lie in $\mathbb{R}^{n \times r}$, sharing the same dimensions. Their corresponding singular value decompositions (SVD) can be given by:

$$Y = \begin{pmatrix} P_1 & P_2 \end{pmatrix} \begin{pmatrix} S_1 & 0 \\ 0 & S_2 \end{pmatrix} \begin{pmatrix} Q_1^\top V_{x1}^H \\ Q_2^H V_{x2}^\top \end{pmatrix}, \tag{2}$$

$$\begin{aligned} X &= U_x \Sigma_x V_x \\ &= \begin{pmatrix} U_{x1} & U_{x2} \end{pmatrix} \begin{pmatrix} \Sigma_{x1} & 0 \\ 0 & 0 \end{pmatrix} \begin{pmatrix} V_{x1} \\ V_{x2} \end{pmatrix}, \end{aligned} \tag{3}$$

where $U_{x1} \in \mathbb{R}^{n_x \times r_x}, U_{x1} \in \mathbb{R}^{n_x \times (n_x - r_x)}, V_{x1} \in \mathbb{R}^{r_x \times r}, V_{x2} \in \mathbb{R}^{(n_x - r_x) \times r}$.

The following theorem states that we cannot directly compute the core features of $X$ from $Y$:

**Theorem 1** *Let $Y$ be the observed data matrix and $X$ the underlying core feature matrix. Then, without additional constraints, there does not exist a pair of matrices $P_1$ and $U_{x1}$ such that $P_1 = U_{x1}$. See Appendix A.2.*

Therefore, we aim to recover the core feature matrix $X$ by representing it as a linear combination of the columns of the observed matrix $Y$. To this end, we introduce a coefficient matrix $H$ and formulate the following least-squares objective:

$$\min_H \|YH - X\|_F^2, \tag{4}$$

where $\|\cdot\|_F$ denotes the Frobenius norm. In this formulation, $X$ is treated as the target (e.g., the ideal low-rank component), and $H$ is the optimization variable that linearly combines the basis vectors in $Y$ to approximate $X$. The optimal $H$ can be derived as:

$$H = \left(Y^\top Y\right)^{-1} Y^\top X. \tag{5}$$

Then, $\hat{X}$, the minimum variance estimate of $X$, can be presented as:

$$\hat{X} = YH = Y \left(Y^\top Y\right)^{-1} Y^\top X, \tag{6}$$

Obviously, we cannot calculate $\hat{\boldsymbol{X}}$ directly by Eq. (6). Note that $\boldsymbol{P_Y} = \boldsymbol{Y}\left(\boldsymbol{Y}^\top\boldsymbol{Y}\right)^{-1}\boldsymbol{Y}^\top$ is the orthogonal projection operator onto the column space of $\boldsymbol{Y}$. Importantly, $\hat{\boldsymbol{X}}$ is denoised when the target $\boldsymbol{X}$ lies entirely within that subspace. To proceed with estimation, we assume that the redundant features in the matrices $\boldsymbol{A}$ and $\boldsymbol{B}$ behave as white noise, i.e., $\boldsymbol{D}^\top\boldsymbol{D} = \sigma_{\boldsymbol{D}}^2 I$, $\boldsymbol{X}^\top\boldsymbol{D} = 0$, where $\sigma_{\boldsymbol{D}}^2$ is the variance of the noise. Then we can simply $\hat{\boldsymbol{X}}$:

$$\hat{\boldsymbol{X}} = \sum_{k=1}^{r_x} \frac{\sigma_k^2 - \sigma_{\boldsymbol{D}}^2}{\sigma_k} \boldsymbol{u}_k \boldsymbol{v}_k^\top, \tag{7}$$

where $\boldsymbol{u}_k$ and $\boldsymbol{v}_k$ is the left and right singular vectors of $\boldsymbol{Y}$, $\sigma_k$ is the $k$-th largest singular value of $\boldsymbol{Y}$ ($\sigma_1 > \sigma_2 > \cdots > \sigma_r$). In traditional signal analysis algorithms, large singular values represent low-frequency data distribution and macro trends, and small singular values represent high-frequency disturbances [25]. However, since $\sigma^2$ in the above formula is unknown, we will define a series of wavelet functions at different scales for feature filtering. Here, we use the heat kernel as a low-pass filter:

$$\phi_{\sigma_j^2, c_j}(\boldsymbol{X}) = \exp\left(-\frac{1}{2\sigma_j^2}||\boldsymbol{X} - c_j||^2\right), \tag{8}$$

where $c_j$ is the center of the $j$-th kernel, $\sigma_j^2$ represents the width of the kernel. By setting a series of different heat kernel widths $\boldsymbol{\sigma^2} = [\sigma_1^2, \sigma_2^2, \cdots]^\top$, defining a series of learnable diagonal matrices $\boldsymbol{g} = [\boldsymbol{g}_1, \boldsymbol{g}_2, \cdots]^\top$ and learnable centers $\boldsymbol{C} = [c_1, c_2, \cdots]^\top$, we can define a wavelet neural network to extract the features of $\hat{\boldsymbol{X}}$ from $\boldsymbol{Y}$:

$$\boldsymbol{H}_{:,i}^{k+1} = \delta\left(\sum_j \phi_{\sigma_j^2, c_j} \boldsymbol{g}_j \phi_{\sigma_j^2, c_j}^{-1} \boldsymbol{H}_{:,j}^k\right), \tag{9}$$

where $\delta(\cdot)$ is the activation function, $\boldsymbol{H}^0 := \boldsymbol{Y}$, and $\hat{\boldsymbol{X}} = \boldsymbol{H}^K$. Here $K$ is the total number of layers and $k \in \{0, 1, \cdots, K-1\}$. As $\phi$ is the heat kernel, i.e. $\phi^{-1}(x) = \phi(-x)$), we could simply Eq. (9) to be:

$$\boldsymbol{H}_{:,i}^{k+1} = \delta\left(\sum_j \phi_{\sigma_j^2, c_j} \boldsymbol{g}_j \phi_{-\sigma_j^2, c_j} \boldsymbol{H}_{:,j}^k\right). \tag{10}$$

Eq. (10) avoids directly calculating the inverse of the function and improves calculation efficiency.

## 3.3 Controlled Knowledge Updating

In the previous section, we derive the core features of the original knowledge. In this section, our goal is to integrate the new knowledge into the LoRA module. The new knowledge can be categorized into two parts: one that does not overlap with the original knowledge, and the other that requires updates due to the outdated nature of the original knowledge. The latter is the primary focus, as it may lead to slight changes in the core features. However, these changes should not alter the overall impact of the original knowledge.

To achieve this goal, we constrain the update of parameters while learning new knowledge through an MLP:

$$\begin{aligned}
\boldsymbol{O}^{k+1} &= \text{MLP}\left(\boldsymbol{H}^{k+1}\right) \\
&= \delta'\left(\boldsymbol{\omega}^{k+1}\boldsymbol{H}^{k+1} + \boldsymbol{b}^{k+1}\right),
\end{aligned} \tag{11}$$

where $\boldsymbol{O}^K$ is the updated matrix $\boldsymbol{A}'$ or $\boldsymbol{B}'$, $\delta'(\cdot)$ is the activation function, $\boldsymbol{\omega}^k \in \boldsymbol{\Omega}$ and $\boldsymbol{b}^k \in \boldsymbol{B}$ are learnable paramters. Defining LoRA-based LLM with pre-trained parameters $\boldsymbol{W}$ and LoRA parameters $\Delta\boldsymbol{W}$ as $\mathcal{A}_{\boldsymbol{W}, \Delta\boldsymbol{W}}(\cdot)$, then the loss function is defined as follows:

$$\begin{aligned}
\mathcal{L}oss = \text{MSE}\left(\mathcal{A}_{\boldsymbol{W}, \Delta\boldsymbol{W}'}(\boldsymbol{Q}), \boldsymbol{G}\right) \\
+ \lambda_1||\boldsymbol{\theta}_1||_a^a + \lambda_2||\boldsymbol{\theta}_2||_b^b,
\end{aligned} \tag{12}$$

where $\Delta\boldsymbol{W}' = \boldsymbol{A}' \times \boldsymbol{B}'^\top$ is parameters of the updated LoRA, $\boldsymbol{G}$ is the ground truth in the new dataset, $\boldsymbol{Q}$ is the input query correspondingly, $\lambda_1$ and $\lambda_2$ are two regularization hyperparameters, $\boldsymbol{\theta}_1 := \{\boldsymbol{g}, \boldsymbol{C}\}$, $\boldsymbol{\theta}_2 := \{\boldsymbol{\Omega}, \boldsymbol{B}\}$ are the learnable parameters in the model, $a$ and $b$

are tworegularization orders which satisfies $a \geq b$. By ensuring that the regularization order for the knowledge retention module's parameters is at least as high as that used for the knowledge update module's regularization term, we can minimize the adjustments made to the retention model's parameters. This, in turn, enhances the overall generalization ability of the model.

## 3.4 Training Procedure

In this subsection, we introduce the training process of DEAL, as shown in Algorithm 1: We begin by locating and obtaining the low-rank matrices $A$ and $B$ in LoRA (lines 1–3). Next, we apply wavelet kernel-based neural networks to extract the core features of historical knowledge from $A$ and $B$ (line 5). Then, another neural network is introduced to superimpose features from new knowledge onto these core features, resulting in a newly constructed low-rank matrix $A'$ or $B'$ (lines 6–11). Using this updated low-rank matrix, we recalculate the LoRA parameters to obtain the fine-tuned model. By calculating the loss with regularization terms and performing back-propagation, we ensure that the model can learn new knowledge while preserving historical features in a controlled manner (lines 13–15).

---

**Algorithm 1** The DEAL framework

**Input:** Pre-trained LoRA-based LLM $\mathcal{A}_{W,\triangle W}$, training samples $\mathcal{D}$
**Output:** Fine-tuned model $\mathcal{A}_{W,\triangle W'}$
 1: $\triangle W' \leftarrow \triangle W$
 2: **for** each batch $(Q, G) \in \mathcal{D}$ **do**
 3:     Extract $A, B$ from $\triangle W'$
 4:     **for** each $Y \in \{A, B\}$ **do**
 5:         $\hat{X} \leftarrow$ Eq. (10), $O \leftarrow$ Eq. (11)
 6:         **if** $Y == A$ **then**
 7:             $A' \leftarrow O$
 8:         **else**
 9:             $B' \leftarrow O$
10:         **end if**
11:     **end for**
12:     $\triangle W' \leftarrow A' \times {B'}^{\top}$
13:     Compute loss $\mathcal{L}oss$ using Eq. (12)
14:     Update parameters via back-propagation
15: **end for**
16: **return** $\mathcal{A}_{W,\triangle W'}$

---

# 4 Experiments

**Datasets.** We evaluate **DEAL** on three continual learning (CL) benchmarks in a sequential task setup, where data from previous tasks is unavailable during training on subsequent ones. These benchmarks are designed to evaluate key challenges in continual learning, including: (i) catastrophic forgetting in semantically related tasks, (ii) inefficient data utilization resulting from restricted access to prior samples, and (iii) scalability across long and diverse task sequences.

**(i) Same-domain tasks:** We use a three-task benchmark consisting of AG News (news classification), DBpedia (entity typing), and Yahoo Answers (question topic prediction). This setup evaluates DEAL's ability to mitigate catastrophic forgetting by retaining transferable knowledge across semantically similar tasks.

**(ii) Domain-shift tasks:** To introduce domain variability, we augment the benchmark with Amazon Reviews [26] for binary sentiment classification. This domain-shift setting assesses generalization under limited data access and distributional shifts, reflecting practical constraints in real-world continual learning.

**(iii) Heterogeneous multi-task learning:** We evaluate on the 15-task benchmark proposed by [27], which spans text classification (AG News, DBpedia, Yahoo, Amazon, Yelp), GLUE tasks (MNLI, QQP, RTE, SST-2) [28], SuperGLUE tasks (WiC, CB, COPA, MultiRC, BoolQ) [29], and IMDB [30]. This benchmark tests DEAL's scalability and robustness across heterogeneous tasks and long task sequences. Full details on dataset preprocessing and prompt construction are provided in Appendices B.1 and E.

**Baselines.** We compare **DEAL** against three LoRA-compatible continual learning baselines:

- **SeqLoRA:** A naive baseline that sequentially updates a single fixed-size LoRA adapter across tasks, without any mechanism to mitigate forgetting.

- **O-LoRA** [31]: A recent method that allocates task-specific adapters to orthogonal subspaces, reducing parameter interference.

- **PerTaskFT:** An oracle baseline in which each task is fine-tuned using a separate LoRA adapter without sharing. Although impractical for deployment, it serves as an upper bound on task-specific retention.

All baselines are evaluated under identical settings, including a fixed adapter architecture, optimization protocol, and tokenization, to ensure fair comparison. Replay-based and non-LoRA methods are excluded to prevent confounding effects from architectural or memory differences. Implementation details are provided in Appendix B.2, and comparisons with broader classes of methods are reported in Appendix G.

**Implementation Details.** We use two pretrained models: **LLaMA 3.1-8B** [32], a decoder-only model fine-tuned on instruction-following corpora, and **T5-Large-Instruct** [33], an encoder-decoder model adapted for general-purpose instruction following. All experiments are conducted on a single NVIDIA A100 GPU. LoRA-based continual fine-tuning is applied to both models using a fixed adapter rank, dropout, learning rate, and batch size across tasks. Gradients are masked as needed to enforce parameter sparsity. Unless stated otherwise, hyperparameters remain consistent across **DEAL** and all baselines. Full training configurations, hardware details, and random seed settings are provided in Appendix B.2.

**Evaluation Metrics.** We report **Average Accuracy (AA)**, the mean test accuracy across all tasks after training concludes, and **ROUGE-1 (R-1)**, which measures unigram F1 overlap between generated outputs and ground-truth labels for free-form generation tasks. Formal metric definitions and application contexts are detailed in Appendix D. We evaluate the training and inference efficiency of DEAL in Appendix F.

## 4.1 Main Results

Table 1 summarizes the continual learning performance of all methods on the three benchmark suites: the 3-task Text Classification (TC), the 4-task Standard CL benchmark, and the 15-task Large-Scale benchmark.

Across all tasks and model backbones, our proposed method, **DEAL**, consistently outperforms both **SeqLoRA** and **O-LoRA**, and achieves performance comparable to the oracle upper bound (PERTASKFT) in terms of average accuracy (AA) and ROUGE-1 F1 (R-1). On the 4-task benchmark with T5-Large, DEAL achieves 78.5% Average Accuracy (AA) and 82.5% ROUGE-1 (R-1), compared to 44.6%/44.6% for SeqLoRA and 71.2%/73.3% for O-LoRA. These improvements stem from DEAL's ability to balance knowledge retention and transfer more effectively. SeqLoRA lacks mechanisms for preserving prior knowledge, leading to severe forgetting. O-LoRA mitigates forgetting via orthogonal subspace constraints but limits beneficial cross-task transfer. In contrast, DEAL integrates shared LoRA modules with dual-branch adapters and stability-aware regularization, enabling robust adaptation while preserving task-specific information.

Table 1: Continual learning performance across three benchmarks.

| Method | 3-Task (TC) | | 4-Task (Standard) | | 15-Task (Large) | |
|---|---|---|---|---|---|---|
| | AA | R-1 | AA | R-1 | AA | R-1 |
| T5 + SeqLoRA | 52.4 | 52.8 | 44.6 | 44.6 | 42.1 | 44.0 |
| T5 + O-LoRA | 85.2 | 87.1 | 71.2 | 73.3 | 70.8 | 80.3 |
| T5 + PerTaskFT | **90.3** | **91.7** | 70.0 | 73.0 | **76.5** | **78.2** |
| T5 + **DEAL(ours)** | 87.7 | 89.3 | **78.5** | **82.5** | 73.9 | 79.1 |
| LLaMA + SeqLoRA | 54.1 | 55.9 | 47.6 | 54.8 | 45.2 | 53.2 |
| LLaMA + O-LoRA | 86.4 | 88.1 | 75.3 | 80.8 | 73.2 | 77.4 |
| LLaMA + PerTaskFT | **88.2** | **90.0** | 77.5 | 79.4 | **77.1** | **82.5** |
| LLaMA + **DEAL(ours)** | 88.9 | 90.2 | **78.9** | **81.3** | 74.6 | 78.9 |

*Note.* Bold indicates the statistically significant improvements
(*i.e.,* two-sided t-test with $p < 0.05$) over the best baseline.

Notably, DEAL approaches the performance of the oracle baseline, **PerTaskFT**, which fine-tunes each task independently without parameter sharing. On the 3-task benchmark with T5-Large, DEAL achieves 87.7% AA, closely matching the 90.3% attained by PerTaskFT. This near-oracle performance is achieved with significantly lower computational overhead. By leveraging modular adapters and regularized updates, DEAL retains task-discriminative signals while benefiting from shared representations, offering a scalable and efficient alternative to task-isolated fine-tuning.

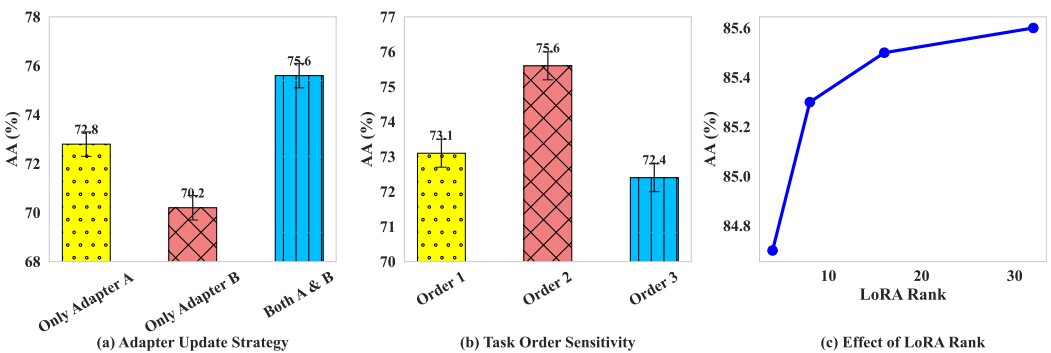

Figure 2: **DEAL** ablations: (a) adapter update strategy, (b) task-order robustness on the 4-task benchmark, (c) LoRA-rank sensitivity on the 3-task benchmark.

The advantages of DEAL become increasingly prominent as task complexity grows. On the 15-task benchmark with LLaMA-3.1-8B, DEAL achieves 74.6% AA, outperforming SeqLoRA by more than 29 percentage points and surpassing O-LoRA as well. In long-horizon continual learning settings, catastrophic forgetting compounds across tasks. DEAL mitigates this degradation through regularization-guided updates and flexible routing, demonstrating strong scalability and robustness to extended task sequences.

## 5    Ablation Studies

We conduct ablation experiments to assess the contribution of five core components in **DEAL**: (i) adapter update strategy, (ii) task-order robustness, (iii) LoRA rank Figure 2, (iv) the kernel function (Table 3), and (v) the regularization strength (Table 4). Unless otherwise specified, all experiments use the T5-Large backbone with fixed random seeds and data orderings.

**Adapter Update Strategy.**   We evaluate three adapter update strategies: updating only Adapter A, updating only Adapter B, and updating both jointly. Joint updates achieve the highest post-task Average Accuracy (AA) of 75.6%, exceeding single-branch updates by 2.8—5.4 percentage points. Across multiple runs, updating only Adapter A consistently outperforms updating only Adapter B. This result reflects the continual learning setting: Adapter A governs global projection directions that capture generalizable semantic patterns and reasoning structures, whereas Adapter B primarily encodes task-specific features. Joint training of both adapters therefore enables the most effective integration of broad knowledge transfer with task specialization.

**Task-Order Robustness.**   To simulate non-stationary task arrivals, we evaluate three random permutations of the 4-task sequence, following the setup in O-LoRA [31]. This experiment aims to examine how sensitive **DEAL** is to variations in task order—a crucial factor in realistic continual learning, where the order of tasks is typically unpredictable. Across the three permutations, the average accuracy (AA) ranges narrowly from 73.1% to 75.6%, reflecting a fluctuation of less than three percentage points. This low variance demonstrates that **DEAL** maintains stable performance regardless of task order, indicating strong robustness and adaptability to dynamic, non-stationary learning environments. The specific task permutations used are detailed in Table 2.

Table 2: Task sequences used for continual learning evaluations.

| Order | Task Sequence |
| --- | --- |
| 1 | DBpedia → Amazon → Yahoo → AG News |
| 2 | DBpedia → Amazon → AG News → Yahoo |
| 3 | Yahoo → Amazon → AG News → DBpedia |

**LoRA Rank.**   Using the 3-task benchmark, we vary the LoRA rank across 4, 8, 16, 32. Accuracy improves significantly from rank 4 to rank 8 (71.5% to 84.3%), then saturates (84.5% at rank 16,

84.6% at rank 32). These results suggest that a compact rank-8 configuration captures most of the task-specific variation while offering strong efficiency in both memory and computation.

**Kernel Functions.** To further support the selection of the heat kernel, we compared it against two alternatives— $f(x) = xe^{-x}$ and quadratic splines—with results summarized in Table 3. All three kernels achieve comparable accuracy, indicating that the model's representational capacity is relatively insensitive to the specific kernel choice. The heat kernel's primary advantage lies in its computational efficiency: it consistently yields orders-of-magnitude reductions in runtime.

This improvement stems from its inverse-free update rule (Eq. (10)), which avoids expensive matrix inversions while maintaining numerical stability. These results demonstrate that the heat kernel offers a favorable trade-off between simplicity and efficiency, supporting its adoption as the default kernel throughout our experiments.

Table 3: Comparison of different kernel functions.

| Kernel Function | Average Accuracy (%) | Training Time (ms/sample) |
|---|---|---|
| $f(x) = xe^{-x}$ | 78.4 | 1759 |
| Quadratic Splines | 78.3 | 1291 |
| **Heat Kernel (Ours)** | **78.5** | **56** |

**Regularization.** We conduct a grid search over regularization weights $(a, b)$, where $a$ regulates the retention branch (wavelet-based) and $b$ governs the adaptation branch (MLP-based). The search ranges over $a \in 1, 5, 10, 20$ and $b \in 1, 2, 5$, capturing a variety of retention-to-adaptation penalty combinations. This ablation aims to identify the optimal trade-off between preserving prior knowledge and facilitating new task adaptation.

Table 4: Grid search over asymmetric regularization weights $(a, b)$.

| $a$ | $b$ | AA (%) |
|---|---|---|
| 1 | 1 | 74.8 |
| 5 | 1 | 83.9 |
| **10** | **2** | **85.5** |
| 10 | 5 | 84.1 |
| 20 | 2 | 82.7 |

*Note.* Bold indicates statistically significant improvements (*i.e.,*, two-sided $t$-test with $p < 0.05$) over the best baseline.

As summarized in Table 4, the best performance is obtained with $(a = 10, b = 2)$, reaching an after-task accuracy (AA) of 85.5%. This configuration outperforms both low-penalty settings (e.g., $(1, 1)$) and heavier regularization schemes (e.g., $(10, 5)$ or $(20, 2)$). These results suggest that appropriately tuning the balance between the retention and adaptation branches is critical: a moderately stronger emphasis on knowledge retention improves stability, while maintaining flexibility in the adaptation pathway supports effective learning of new tasks. Notably, this balance mitigates forgetting without hindering new knowledge acquisition, highlighting the importance of carefully calibrated regularization in continual learning.

## 6 Related Work

**Continual Learning for Large Language Models** Continual learning (CL) with large language models (LLMs) presents a fundamental challenge: acquiring new knowledge without catastrophic forgetting, while ensuring efficiency and adaptability. Existing approaches largely fall into three categories: memory-based, regularization-based, and subspace isolation-based methods.

Memory-based approaches, such as experience replay (ER) [34], mitigate forgetting by revisiting buffered past data. However, these methods raise concerns about scalability and data privacy, limiting their practicality in real-world LLM deployments. Regularization-based methods constrain parameter updates to preserve previously learned knowledge. For instance, CLoRA [8] introduces angular regularization between task-specific LoRA adapters. While computationally lightweight, such approaches often under perform on dissimilar tasks due to overly restrictive adaptation dynamics. Subspace-based techniques offer a memory-free alternative by decoupling task representations. O-LoRA [31], for example, updates LoRA parameters within orthogonal subspaces of prior tasks, effectively reducing representational interference. However, it faces two key limitations: (i) orthogonality is enforced only in first-order gradient space, overlooking higher-order interactions; and (ii) all subspace directions are treated uniformly, with no prioritization of semantically meaningful components. The TRACE

benchmark [35] further highlights trade-offs in CL for LLMs: while full-parameter fine-tuning yields high per-task accuracy, it suffers from severe forgetting; in contrast, naive LoRA tuning preserves general capabilities but degrades instruction-following performance.

In contrast to prior work, our method introduces structural regularization and selective routing, aligning task-specific updates with semantically salient directions while preserving cross-task generalization. This design promotes transferability and robustness without relying on external memory or imposing rigid parameter constraints.

**Parameter-Efficient Tuning**    Parameter efficient tuning (PET) approaches, such as adapters [36–38], prompt tuning [39, 40], and LoRA [6, 41], reduce training cost by updating only a small subset of model parameters. Several extensions of LoRA have been proposed to improve adaptability and representational capacity. ReLoRA[42] enables high-rank representations by scheduling low-rank updates across training phases. FLORA[43] employs stochastic resampling to approximate richer adaptation structures with low memory overhead. To support knowledge transfer across tasks, modular PET strategies have also been developed. LoraHub[44] assembles reusable task-specific adapters that can be composed dynamically. MOLE[45] treats multiple LoRA modules as experts and selects among them using a learned gating mechanism.

In contrast to prior work that focuses on isolated improvements in capacity or flexibility, we propose a wavelet regularized continual tuning framework that jointly addresses knowledge retention and adaptive transfer. Our method sustains performance across evolving tasks without relying on data replay, while incurring minimal computational overhead.

# 7    Conclusion

We present **DEAL**, a continual learning framework that integrates instruction-guided fine-tuning with lightweight adapter updates and structured regularization. Built on a LoRA-style architecture, **DEAL** enables scalable, interference-resistant learning across sequential tasks while maintaining strong performance on diverse benchmarks. Extensive experiments highlight three core advantages:

- **Reduced Forgetting.** DEAL employs regularized low-rank updates to preserve task-relevant subspaces, mitigating catastrophic forgetting without relying on explicit memory buffers.

- **Efficient Adaptation.** Shared LoRA components serve as inductive priors, accelerating convergence and supporting efficient generalization in low-resource scenarios.

- **Scalability.** By introducing only a small number of task-specific parameters, DEAL scales effectively to long task sequences and large LLM backbones such as LLaMA-3.1-8B.

**Limitations and Future Directions.**    While DEAL demonstrates strong performance in diverse continual learning scenarios, it currently assumes a fixed task order and static model capacity. Promising directions include lightweight rehearsal mechanisms, dynamic capacity allocation, and enhancing forward transfer through meta-regularization. Addressing robustness under ambiguous task boundaries remains an open challenge.

# Acknowledgements

This work was partially supported by the National Natural Science Foundation of China (No. 62502404), the Research Impact Fund of the Hong Kong Research Grants Council (No. R1015-23), the Collaborative Research Fund of the Research Grants Council (No. C1043-24GF), the General Research Fund of the Research Grants Council (No. 11218325), and the Institute of Digital Medicine at City University of Hong Kong (No. 9229503). Additional support was provided by Huawei (Huawei Innovation Research Program), Tencent (CCF–Tencent Open Fund; Tencent Rhino-Bird Focused Research Program), Alibaba (CCF–Alimama Tech Kangaroo Fund No. 2024002), Didi (CCF–Didi Gaia Scholars Research Fund), Kuaishou, and ByteDance.

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

# Appendix

## A  Supplementary Algorithmic Analysis

### A.1  Notation

Let $\boldsymbol{X} \in \mathbb{R}^{n_x \times r}$ be a matrix of rank $r_x$. Its singular value decomposition (SVD) is:

$$
\begin{aligned}
\boldsymbol{X} &= \boldsymbol{U}_x \boldsymbol{\Sigma}_x \boldsymbol{V}_x^\top \\
&= [\boldsymbol{U}_{x1} \ \boldsymbol{U}_{x2}] \begin{bmatrix} \boldsymbol{\Sigma}_{x1} & 0 \\ 0 & 0 \end{bmatrix} \begin{bmatrix} \boldsymbol{V}_{x1}^\top \\ \boldsymbol{V}_{x2}^\top \end{bmatrix}.
\end{aligned}
\tag{13}
$$

Here, $\boldsymbol{U}_{x1} \in \mathbb{R}^{n_x \times r_x}$ and $\boldsymbol{V}_{x1} \in \mathbb{R}^{r \times r_x}$ correspond to the principal subspace, while $\boldsymbol{U}_{x2}$ and $\boldsymbol{V}_{x2}$ span the orthogonal complement.

### A.2  Proof of Theorem 1

Consider a perturbation $\boldsymbol{D} \in \mathbb{R}^{n_x \times r}$, and define:

$$
\boldsymbol{Y} = \boldsymbol{X} + \boldsymbol{D}.
\tag{14}
$$

We decompose $\boldsymbol{D}$ via projection onto the column spaces of $\boldsymbol{V}_{x1}$ and $\boldsymbol{V}_{x2}$:

$$
\begin{aligned}
\boldsymbol{Y} &= \boldsymbol{X} + \boldsymbol{D} \left( \boldsymbol{V}_{x1} \boldsymbol{V}_{x1}^\top + \boldsymbol{V}_{x2} \boldsymbol{V}_{x2}^\top \right) \\
&= \left( \boldsymbol{X} \boldsymbol{V}_{x1} + \boldsymbol{D} \boldsymbol{V}_{x1} \right) \boldsymbol{V}_{x1}^\top + \left( \boldsymbol{D} \boldsymbol{V}_{x2} \right) \boldsymbol{V}_{x2}^\top.
\end{aligned}
\tag{15}
$$

Denoting SVDs of each term:

$$
\boldsymbol{X} \boldsymbol{V}_{x1} + \boldsymbol{D} \boldsymbol{V}_{x1} = \boldsymbol{P}_1 \boldsymbol{S}_1 \boldsymbol{Q}_1^\top
\tag{16}
$$

$$
\boldsymbol{D} \boldsymbol{V}_{x2} = \boldsymbol{P}_2 \boldsymbol{S}_2 \boldsymbol{Q}_2^\top,
\tag{17}
$$

with $\boldsymbol{P}_1^\top \boldsymbol{P}_2 = 0$, we can express $\boldsymbol{Y}$ as:

$$
\boldsymbol{Y} = [\boldsymbol{P}_1 \ \boldsymbol{P}_2] \begin{bmatrix} \boldsymbol{S}_1 & 0 \\ 0 & \boldsymbol{S}_2 \end{bmatrix} \begin{bmatrix} \boldsymbol{Q}_1^\top \boldsymbol{V}_{x1}^\top \\ \boldsymbol{Q}_2^\top \boldsymbol{V}_{x2}^\top \end{bmatrix}.
\tag{18}
$$

This constitutes the SVD of $\boldsymbol{Y}$, and the basis $\boldsymbol{P}_1$ generally differs from $\boldsymbol{U}_{x1}$ due to perturbation. This completes the proof.

## B  Experimental Setup

### B.1  Datasets

Table 5 summarizes all datasets used across the continual learning benchmarks. These datasets were also employed in O-LoRA [31], where each is framed as a classification task using a unified instruction-based text-to-text format.

Table 5: Overview of datasets used in experiments.

| # | Dataset | Source | Task Type | Evaluation Metric |
|---|---------|--------|-----------|-------------------|
| 1 | AG News | CL Benchmark | Topic Classification | Accuracy |
| 2 | DBpedia | CL Benchmark | Entity Typing | Accuracy |
| 3 | Yahoo | CL Benchmark | Topic Classification | Accuracy |
| 4 | Amazon | CL Benchmark | Sentiment Analysis | Accuracy |
| 5 | MNLI | GLUE | Natural Language Inference | Accuracy |
| 6 | QQP | GLUE | Paraphrase Detection | Accuracy |
| 7 | RTE | GLUE | Natural Language Inference | Accuracy |
| 8 | SST-2 | GLUE | Sentiment Analysis | Accuracy |
| 9 | WiC | SuperGLUE | Word Sense Disambiguation | Accuracy |
| 10 | CB | SuperGLUE | Natural Language Inference | Accuracy |
| 11 | COPA | SuperGLUE | Causal Reasoning | Accuracy |
| 12 | BoolQ | SuperGLUE | Boolean QA | Accuracy |
| 13 | MultiRC | SuperGLUE | Multi-hop QA | Accuracy |
| 14 | IMDB | External | Sentiment Analysis | Accuracy |

## B.2 Implementation Details

We implement **DEAL** using the Hugging Face Transformers library and perform training with FP16 mixed precision on a single NVIDIA A100 GPU. Unless otherwise stated, we adopt a consistent experimental setup across both LLaMA-3.1 and T5-large backbones, tailored for low-resource, instruction-driven continual learning. All models are trained with adapter-based fine-tuning using LoRA, where we set the rank $r = 32$ for LLaMA and $r = 16$ for T5, selected based on backbone capacity. Optimization is performed using AdamW with a constant learning rate scheduler. Regularization is enforced via $\ell_p$-norm constraints on adapter weights and MLP modules, with details provided in Table 6.

Table 6: Hyperparameter settings for **DEAL** on LLaMA-3.1 and T5-large.

| Hyperparameter | LLaMA-3.1 | T5-large |
|----------------|-----------|----------|
| LoRA Rank $r$ | 32 | 16 |
| Learning Rate | 1e-5 | 1e-5 |
| Batch Size | 8 | 4 |
| Gradient Accum. Steps | 4 | 2 |
| Epochs | 1 | 1 |
| Max Source Length | 512 | 512 |
| Max Target Length | 50 | 50 |
| Generation Max Length | 50 | 50 |
| Warmup Steps | 0 | 0 |
| Dropout | – | – |
| Optimizer | AdamW | AdamW |
| Scheduler Type | constant | constant |
| Regularization $\lambda_1$ | 0.01 | 0.01 |
| Regularization $\lambda_2$ | 0.001 | 0.001 |
| $\|\theta\|_5$ Norm Reg. | ✓ | ✓ |
| $\|MLP\|_2$ Norm Reg. | ✓ | ✓ |
| LoRA Modules | q_proj, v_proj | q, v |
| Task Type | CausalLM | Seq2SeqLM |

We set $\lambda_1 = 0.01$ to constrain the $\ell_5$-norm of adapter parameters $\theta$, and $\lambda_2 = 0.001$ to regularize the $\ell_2$-norm of task-specific MLP weights, when applicable. All experiments are conducted for one epoch over instruction-based task mixtures, without early stopping or checkpointing unless explicitly mentioned.

# C  Case Study Examples

---

**Case 1: DBpedia Classification Task**

**Prompt:**
You are a smart AI evaluator. Given an input paragraph and three model outputs *(Base Model, Adapter after DBpedia, Adapter after DBpedia→Amazon)*, judge which output(s) correctly classify the paragraph according to the true label.

**Question:**
What is the topic of the following paragraph? Choose one from the options below.
["Company", "Educational Institution", "Artist", "Athlete", "Office Holder", "Mean of Transportation", "Building", "Natural Place", "Village", "Animal", "Plant", "Album", "Film", "Written Work"]

**Input (DBpedia sample)**

**Label:** Office Holder

**Text:** Raimundas Palaitis (born 23 October 1957) is a Lithuanian politician. He was Minister of the Interior from 2008 to 2012.

**Base Model:** Building

**Adapter after DBpedia:** Office Holder

**Adapter after DBpedia→Amazon:** Office Holder

- - - - - - - - - - - - - - - - - - - - - - - - - - - - - - - - - - - - - - - - - - - - -

**Expected Answer:**
The base model incorrectly predicted "Building," which is unrelated to the paragraph. In contrast, both adapters correctly predicted "Office Holder," indicating that they effectively learned the DBpedia task and that the second adapter retained this knowledge even after subsequent training on the Amazon task

---

**Case 2: Amazon Sentiment Task**

**Prompt:**
Given an input review and three model outputs, decide whether the sentiment is classified correctly and briefly explain why.

**Question:**
What is the sentiment of the following review? Choose one from
["very negative", "negative", "neutral", "positive", "very positive"]

**Input (Amazon sample)**

**Label:** very negative

**Text:** I don't understand how they can advertise that this humidifier can work up to 12 hours. It runs out of water so fast. I have to get up in the middle of the night and refill it. I would say it works well for about 2 hours. So, while it is inexpensive, you get what you pay for.

**Base Model:** neutral

**Adapter after DBpedia:** neutral

**Adapter after DBpedia→Amazon:** very negative

- - - - - - - - - - - - - - - - - - - - - - - - - - - - - - - - - - - - - - - - - - - - -

**Expected Answer:**
Both the base model and the first adapter misclassified the review as "neutral". Only the adapter further fine-tuned on Amazon data predicted the correct label, "very negative," demonstrating its ability to acquire new task knowledge that the earlier models failed to capture.

Figure 3: Two case studies demonstrating continual learning and knowledge retention across classification tasks.

## D  Evaluation Metrics

We adopt the following standard metrics for continual learning evaluation:

- **Average Accuracy (AA)**: Measures the average test accuracy across all tasks after the final task is learned:

$$AA = \frac{1}{T} \sum_{i=1}^{T} a_{i,T},$$

  where $a_{i,T}$ is the test accuracy on task $i$ after training on task $T$.
- **ROUGE-1 (R-1)**: Used for generative label decoding, computed as the unigram $F_1$ between model output and reference:

$$R-1 = \frac{2 \cdot P \cdot R}{P + R}, \quad P = \frac{|y \cap y^\star|}{|y|}, \quad R = \frac{|y \cap y^\star|}{|y^\star|}.$$

## E  Instruction Prompts

We adopt the task-specific instruction prompts introduced in O-LoRA [31], as summarized in Table 7.

Table 7: Instruction prompts provided to the model for each task.

| Task | Prompt |
|---|---|
| NLI | *What is the logical relationship between "sentence 1" and "sentence 2"? Choose one from the options.* |
| QQP | *Do "sentence 1" and "sentence 2" express the same meaning? Choose one from the options.* |
| SC | *What is the sentiment of the following passage? Choose one from the options.* |
| TC | *What is the topic of the following passage? Choose one from the options.* |
| BoolQA | *According to the passage, is the statement true or false? Choose one from the options.* |
| MultiRC | *Based on the passage and question, is the candidate's answer correct? Choose one from the options.* |
| WiC | *Given a word and two sentences, is the word used with the same sense in both? Choose one from the options.* |

## F  Training and Inference Efficiency

**Training Efficiency.**  We measure the training throughput and GPU memory usage on the DBpedia dataset with the T5-large backbone (Table 8).

Table 8: Training efficiency on DBpedia with T5-large.

| Method | Training Throughput (samples/sec) | GPU Mem. Train (GB) |
|---|---|---|
| LoRA | 31.62 | 20.41 |
| DEAL | 17.88 | 22.93 |

**Inference Efficiency.**  We evaluate inference latency and GPU memory usage under the same setting (Table 9).

Table 9: Inference efficiency on DBpedia with T5-large.

| Method | Inference Latency (ms/sample) | GPU Mem. Infer (GB) |
|---|---|---|
| LoRA | 71.89 | 3.15 |
| DEAL | 73.32 | 3.16 |

**Discussion.**  As shown in Tables 8 and 9, DEAL incurs a ~43% drop in training throughput and a small increase in training GPU memory, while inference-time performance remains nearly identical to LoRA in both latency and memory usage. Since the wavelet module is only active during training and enables consistent performance gains (see Table 1), this overhead represents a worthwhile trade-off for better generalization and robustness.

# G Comparison with Additional Continual Learning Methods

To provide a comprehensive evaluation, we expand our comparisons to include a wide range of strong non-LoRA continual learning (CL) baselines. Table 10 reports the average accuracy (AA) on a standard CL benchmark using the T5-large backbone.

Our method, DEAL, achieves the highest average accuracy while maintaining parameter efficiency. It outperforms both rehearsal-based approaches (Replay) and regularization-based techniques (EWC, LwF). Additionally, DEAL surpasses prompt-based strategies (L2P, ProgPrompt) and recent competitive baselines (LFPT5, LB-CL), highlighting its effectiveness as a general-purpose solution for continual learning.

Table 10: Comparison with Additional Continual Learning Methods.

| Method | Average Accuracy (AA) |
|---|---|
| SeqSVD | 63.3 |
| Replay | 52.0 |
| EWC | 45.3 |
| LwF | 52.9 |
| L2P | 60.5 |
| LFPT5 | 71.2 |
| ProgPrompt | 76.0 |
| LB-CL | 76.5 |
| **DEAL (Ours)** | **78.5** |

- **SeqSVD**: Learns a fixed-size SVD parameter space across sequential tasks without regularization or replay.
- **Replay**: Rehearses past samples to prevent forgetting by storing and replaying them during training [46].
- **EWC**: Mitigates forgetting by penalizing changes to important parameters estimated via Fisher information [47].
- **LwF**: Preserves knowledge of past tasks through knowledge distillation without storing old data [48].
- **L2P**: Introduces learnable prompts for continual learning to guide pre-trained models effectively [49].
- **LFPT5**: Proposes a unified framework for lifelong few-shot learning based on prompt tuning of T5 [50].
- **ProgPrompt**: Designs progressive prompts to adapt large language models for continual learning [51].
- **LB-CL**: Achieves parameter-efficient continual learning by learning more but disturbing less [52].

