# OpenReview forum: "Data Efficient Adaptation in Large Language Models via Continuous Low-Rank Fine-Tuning"
_NeurIPS.cc/2025/Conference — NeurIPS 2025 poster_

### Official Review · Reviewer_naB7 · 2025-06-27

**Clarity:** 3
**Significance:** 3
**Originality:** 3
**Rating:** 4
**Confidence:** 4

**Summary:**

This paper presents DEAL, a continual learning framework for LoRA-based large language models that addresses catastrophic forgetting through two key components: a wavelet kernel-based module that preserves core historical knowledge features, and a controlled updating module that integrates new information via asymmetric regularization. Experiments show DEAL consistently outperforms existing baselines while maintaining computational efficiency and achieving near-oracle performance without additional inference overhead.

**Questions:**

1. Why did you choose heat kernels specifically? Have you experimented with other types of wavelets
2. Can you provide detailed timing comparisons during training? The wavelet neural network introduces computation overhead - at what point does this overhead outweigh the benefits?

**Ethical Concerns:**

["NO or VERY MINOR ethics concerns only"]

**Final Justification:**

The responses from the authors of this paper addressed most of my concerns. Considering the contribution and application potential of this paper, I decided to keep my positive score.

**Limitations:**

1. Limited baseline scope: The authors only compare against LoRA-compatible methods but fail to acknowledge or discuss the absence of comparisons with established continual learning approaches such as experience replay, elastic weight consolidation.
2. The paper does not address how the approach handles very long task sequences or situations where new knowledge fundamentally conflicts with historical knowledge. This is particularly concerning for real-world deployments where models may encounter hundreds of tasks over time or face contradictory information.

**Paper Formatting Concerns:**

No major formatting issues

**Quality:**

3

**Strengths And Weaknesses:**

Strengths

1. Provides mathematical proof showing the infeasibility of direct feature extraction from singular LoRA matrices, which motivates the wavelet kernel approach.
2. Comprehensive evaluation: Extensive experiments spanning multiple domains (text classification, GLUE, SuperGLUE) demonstrate consistent improvements.
3. No additional inference overhead since updated LoRA matrices directly replace original ones, making it deployment-friendly.

Weaknesses

1. Only compares against LoRA-compatible methods, missing comparisons with other continual learning approaches.
2. Unclear how the approach scales to very long task sequences or when dealing with conflicting knowledge updates over time.

---

> ### Author Rebuttal · Authors · 2025-07-31
>
> We appreciate the valuable suggestions and your recognition of the solid techniques and the novelty of the paper, which is encouraging.
>
> ---
>
> **Q1: Why did you choose heat kernels specifically? Have you experimented with other types of wavelets?**
>
> A1: Thank you for the question. The purpose of selecting heat kernels is to reduce the computational complexity of the proposed model. The concern arises primarily due to the inverse function $\phi^{-1}$ appearing in Equation (10) of our method (line 151-152, page 5). In general, computing the inverse of a nonlinear function can be computationally expensive. However, for the specific case where we use the heat kernel $h(x) = e^{-x}$, its inverse satisfies $h^{-1}(x) = h(-x)$, which is a simple negation. This property eliminates the need for explicit inverse computation, significantly improving computational efficiency in practice.
>
> For reference, our update step is as follows:
>
> $$
> \boldsymbol{H} _{:,i}^{k+1} = \delta\left( \sum _j \phi _{\sigma _j^2, c _j} \ \boldsymbol{g} _j \ \phi _{-\sigma _j^2, c _j} \boldsymbol{H} _{:,j}^k \right).
> $$
>
> This **inverse-free formulation** is the main reason we adopt the exponential (heat kernel) form for \(\phi\), as it enables both conceptual simplicity and computational efficiency. We appreciate the reviewer’s insightful comment.
>
> In addition, we have also compared two different types of functions as kernels: $f(x) = x e^{-x}$ and quadratic splines. The experimental results are shown in the table below. As observed, when using the heat kernel, the model incurs less computational overhead while achieving similar accuracy.
>
> |Method|Average Accuracy|Traning Time (ms/per sample)|
> |:--|:--|:--|
> |$f(x) = xe^{-x}$|78.4|1,759|
> |Quadratic Splines|78.3|1,291|
> |**Ours**|**78.5**|**56**|
>
> ---
>
> **Q2: Can you provide detailed timing comparisons during training? The wavelet neural network introduces computation overhead – at what point does this overhead outweigh the benefits?**
>
> A2: Thank you for the question. We summarize the timing efficiency and overhead of our method compared to LoRA as follows:
>
> - **Training Efficiency**: Our method introduces a modest increase in training time due to the wavelet module, which processes input features in the frequency domain. However, this additional overhead remains limited and does not hinder scalability.
> - **Inference Efficiency**: The wavelet module is **completely discarded during inference**. Instead, the trained output of the module (LoRA A or B) is directly substituted with the parameters in the meta-model. As a result, there is **no additional** latency or memory usage during inference compared to the base model or LoRA.
>
> To concretely quantify this, we evaluated our method on the DBpedia dataset using the T5-large backbone. The results are as follows:
>
> | Method | Training Throughput (samples/sec) | GPU Memory during Training (GB) | Inference Latency (ms/sample) | GPU Memory during Inference (GB) |
> |--------|-----------------------------------|----------------------------------|-------------------------------|----------------------------------|
> | LoRA   | 31.62                           | 20.41                             | 71.89                         | 3.15                             |
> | DEAL   | 17.88                           | 22.93                             | 73.32                         | 3.16                             |
>
> As shown, **DEAL introduces a ~43% drop in training throughput** and a *slight increase in GPU memory during training*, but the **inference-time performance remains nearly identical to that of LoRA** in both latency and memory consumption.
>
> Given that the wavelet module is only active during training and enables consistent performance improvements(as shown in Table1), we believe this overhead is a worthwhile trade-off for enhanced model generalization and robustness.
>
> ---
>
> **W1: Only compares against LoRA-compatible methods, missing comparisons with other continual learning approaches.**
>
> A3: Thanks for your suggestion. We have expanded our experimental comparisons to include a broader range of strong baselines. The following table reports the average accuracy (AA) on the standard Continual Learning (CL) benchmark using the T5-large backbone:
>
> | Method      | Average Accuracy (AA) |
> |-------------|------------------------|
> | IncLoRA     | 62.2                   |
> | SeqSVD      | 63.3                   |
> | Replay      | 52.0                   |
> | EWC         | 45.3                   |
> | LwF         | 52.9                   |
> | L2P         | 60.5                   |
> | LFPT5       | 71.2                   |
> | ProgPrompt  | 76.0                   |
> | LB-CL       | 76.5                   |
> | **DEAL**    | **78.5**               |
>
> **Method Descriptions:**
>
> - **IncLoRA**: Incremental learning of new LoRA parameters over a sequence of tasks, without any regularization or replay of prior data.
> - **SeqSVD**: Learning a fixed-size SVD parameter space across sequential tasks, without regularization or replay.
> - **Replay**: _D. Lopez-Paz and M. Ranzato, "Gradient episodic memory for continual learning," NeurIPS 2017._
> - **EWC**: _J. Kirkpatrick et al., "Overcoming catastrophic forgetting in neural networks," PNAS 2017._
> - **LwF**:  _Z. Li and D. Hoiem, "Learning without forgetting," ECCV 2016._
> - **L2P**: _X. Wang et al., "Learning to prompt for continual learning," CVPR 2022._
> - **LFPT5**: _H. Qin et al., "LFPT5: A unified framework for lifelong few-shot language learning based on prompt tuning of T5," NeurIPS 2022._
> - **ProgPrompt**:  _L. Wang et al., "ProgPrompt: Continual learning for language models," ICLR 2023._
> - **LB-CL**: _F. Qiao and M. Mahdavi, "Learn more, but bother less: Parameter-efficient continual learning," *NeurIPS*, 2024._
>
> ---
>
> **W2: Unclear how the approach scales to very long task sequences or when dealing with conflicting knowledge updates over time.**
>
> **A4:** Thank you for the insightful comment. We agree that scaling to very long task sequences and handling conflicting updates over time is a central challenge in continual learning. In our work, we have already evaluated the method on a relatively demanding benchmark with **15 sequential tasks** (lines 187–192, page 6), as shown in **Table 1**.
>
> It is worth noting that, as shown in Table 1, **almost all continual learning methods perform significantly worse than PerTaskFT**, suggesting that scaling to a large number of tasks remains inherently difficult for the entire field—not just our method.
>
> Nevertheless, our approach maintains competitive performance across all task counts and shows less degradation than other baselines as the number of tasks increases. We believe this demonstrates the scalability potential of our method, and we plan to explore even longer task sequences in future work.

---

> > ### Comment · Reviewer_naB7 · 2025-08-04
> >
> > Thanks to the author for the response, which addressed most of my concerns. I will keep my positive score.

---

### Official Review · Reviewer_A3Fs · 2025-07-03

**Clarity:** 1
**Significance:** 2
**Originality:** 2
**Rating:** 2
**Confidence:** 3

**Summary:**

The work introduces a continual learning framework that uses small amounts of new data for continuous learning, thereby avoiding the need for relearning. The method uses a wavelet kernel to preserve historical knowledge and deploy differentiated regularization terms to control the knowledge updating process. Experimental results shows the effectiveness of the proposed framework to maintain high performance across different tasks.

**Questions:**

I have some questions about different parts of the paper, which are listed in the following:

1. In line 3 and 14, what is meant by "privacy-sensitive scenarios" and "data privacy"? In the main body of the paper, the authors have not specifically discussed data privacy. Does it mean the data privacy of the fine-tuning data? If so, why is it important in the context of continual learning?

2. Line 43: there is a typo: One -> On

3. Line 123: What is meant by "core feature"? and why can it be found by SVD? This part and the few lines after that really need to be clarified.

4. Line 125: The dimensions of Y and X should be the same, right?

5. In Theorem 1: Why should $P_1$ be equal to $U_{x_1}$? As mentioned earlier, this whole section needs to be re-written and clarified.

6. Line 133: The optimization problem (4) assumes $X$ is given and the minimization is over $Z$, right? What is $f$?

7. Line 134: Is it a typo? $H$ should change to $Z$?

8. Line 135: $\hat{X}$ is the estimate of $X$, but $X$ appears in equation (6)! How is it possible?

9. Line 136: The expression for $YZ$ is different from that in equation (6).

10. The notations in eq. 8 and 9 for the function $\phi$ are different.

11. Line 16 of algorithm 1: What is the "$=0$" at the end?

12. In Table 1, and "4-Task (Standard)" column, why is DEAL better than the oracle PerTaskFT? The oracle one should perform better right? because it uses a separate LoRA adapter for each task and there is zero sharing.

13. Line 251: This line refers to Fig. 2 a. Why does updating only the "task-specific adapter B" results in a better average accuracy (AA) than when updating only "the shared adapter A"? Shouldn't the task-speciifc one be better? Updating B adapts the model to the new upcoming tasks, right?

**Ethical Concerns:**

["NO or VERY MINOR ethics concerns only"]

**Final Justification:**

The authors have acknowledged that **"the work (F. Qiao et al., Learn more, but bother less: Parameter Efficient Continual Learning, NeurIPS 2024) serves as a strong and directly relevant baseline"**, but have not used it as a baseline. This work is a strong baseline from Neurips 2024 that should have been used as a baseline. Without complete comparison with this strong and recent work, I strongly doubt about the contribution of this work in terms of improving the SoTA. I strongly believe that, despite being interesting, this work should be improved (in its writing and experiments) and include the above baseline in its experiments to prove its contribution. Only then, it can be resubmitted to the next venues.

Therefore, I keep my score and also encourage the authors to improve their draft as explained.

**Limitations:**

The writing of the paper really needs to be improved to make it more understandable and easier to follow. The related works section can be pushed to the appendix and can be replaced with some background about wavelet filtering and other details related to the proposed idea. Also, the baselines mentioned above need to be discussed and compared with if they are related. It seems that they are all closely related. Overall, the paper and the results need to be improved.

**Paper Formatting Concerns:**

no concerns.

**Quality:**

1

**Strengths And Weaknesses:**

Strengths:
1. The considered topic of continual learning of LLMs is important and interesting.
2. Experimental results have been reported for three different benchmarks.

Weaknesses:
1. The paper is not well-written and is hard to follow, especially the whole section 3. It seems that there are multiple typos in there as well. See below. The writing of the paper needs to be improved to clarify further about the proposed idea and make it understandable.

2. Some directly related baselines are not studied. See the following works, which are very closely related. Comparing with them or at least discussing them and their potential shortcomings is essential.

[1] M. Wistuba, et. al., "Continual Learning with Low Rank Adaptation", Neurips 2023.

[2] Y. Liang, et. al., "InfLoRA: Interference-Free Low-Rank Adaptation for Continual Learning", CVPR 2024.

[3] F. Qiao, et. al., "Learn more, but bother less: parameter efficient continual learning", Neurips 2024.

[4] X. Wei, et. al., "Online-LoRA: Task-free Online Continual Learning via Low Rank Adaptation", WACV 2024.

3. The section 6 (related work) can be put at the beginning of the paper (e.g. after section 1) to provide a better view of the content to the reader.

4. The experimental results are limited due to not comparing to multiple baselines. Currently, from the three considered baselines, one is a naive one and one is an oracle method.

---

> ### Author Rebuttal · Authors · 2025-07-31
>
> Thanks for your comments, and we would like to clarify your major concerns:
>
> ---
>
> **Q1: In line 3 and 14, what is meant by "privacy-sensitive scenarios" and "data privacy"? In the main body of the paper, the authors have not specifically discussed data privacy. Does it mean the data privacy of the fine-tuning data? If so, why is it important in the context of continual learning?**
>
> **A1:** Thank you for the question. We will clarify it in the revised manuscript to avoid misunderstandings. The private information in the data samples is not helpful for the target learning task in continuous learning and is considered as noise data. We use this example to highlight that continual learning methods are often prone to catastrophic forgetting and poor model performance in such settings, as illustrated in lines 5-6. We apologize for the misunderstanding and will delete the potentially misleading description in the revised manuscript.
>
> ---
>
> **Q2-Q9: Paper clarity issues in lines 121-149.**
>
> Thanks for your questions. In the revised version, we will improve the clarity of this part and correct any typos. There do have some misunderstandings: For example, the Question 5 is "Why should $P_1$ be equal to $U_{x_1}$?", while in lines 129-131, Theorem 1, we said "... there ***does not*** exist a pair of matrices $P_1$ and $U_{x_1}$ such that $P_1=U_{x_1}$". In addition, as for Question 6, $X$ is the variable that we need to estimate rather than it is given, as we illustrated in line 132: "... to find the optimal estimate of $X$ ...".
>
> We apologize once again for the misunderstanding. We have reorganized lines 121-149 and corrected the typos you pointed out to address your concerns, as detailed below:
>
> In the LoRA module, the matrices $A$ and $B$ are singular, meaning they are not full-rank. This presents a challenge for effective feature extraction. We assume that the singular matrix $Y := A$ or $B$ can be decomposed into a task-relevant component $X$ and a redundant or noisy component $D$, i.e., $Y = X + D$.
>
> Here, $X$ denotes the core feature matrix, capturing the intrinsic low-rank structure that encodes task-relevant semantics. According to the Eckart–Young–Mirsky theorem (Eckart and Young, 1936), the best low-rank approximation of a matrix in the Frobenius norm sense is achieved via truncation of its singular value decomposition (SVD). This motivates our use of truncated SVD to estimate $X$ from the observed matrix $Y$, with the goal of recovering task-relevant features from its singular representation.
>
> We further assume that both $Y$ and $X$ lie in $\mathbb{R}^{n \times r}$, sharing the same dimensions, since $Y$ is formed by the additive composition of $X$ and the residual term $D$. Their corresponding singular value decompositions (SVD) are given by:
>
> $$
> Y =
> \begin{pmatrix}
> P_1 & P_2
> \end{pmatrix}
> \begin{pmatrix}
> S_1 & 0 \\\\
> 0 & S_2
> \end{pmatrix}
> \begin{pmatrix}
> Q_1^\top V_{x1}^H \\\\
> Q_2^H V_{x2}^\top
> \end{pmatrix},
> \tag{2}
> $$
>
> $$
> \begin{align}
> X &= U_x \Sigma_x V_x \\\\
> &=
> \begin{pmatrix}
> U_{x1} & U_{x2}
> \end{pmatrix}
> \begin{pmatrix}
> \Sigma_{x1} & 0 \\\\
> 0 & 0
> \end{pmatrix}
> \begin{pmatrix}
> V_{x1} \\\\
> V_{x2}
> \end{pmatrix},
> \tag{3}
> \end{align}
> $$
>
> where $U _{x1} \in \mathbb{R}^{n _x \times r _x}$, $U _{x1} \in \mathbb{R}^{n _x \times (n _x-r _x)}$, $V _{x 1} \in \mathbb{R}^{r _x \times r}$, $V _{x 2} \in \mathbb{R}^{(n _x-r _x) \times r}$.
>
> The following theorem states that we cannot directly compute the core features of $X$ from $Y$:
>
> **Theorem 1.**
> Let $Y$ be the observed data matrix and $X$ the underlying core feature matrix. Then, without additional constraints, there does not exist a pair of matrices $P_1$ and $U_{x1}$ such that $P_1 = U_{x1}$. (See Appendix A.2)
>
> Therefore, we aim to recover the core feature matrix $X$ by representing it as a linear combination of the columns of the observed matrix $Y$. To this end, we introduce a coefficient matrix $H$ and formulate the following least-squares objective:
>
> $$
> \min_{H} \| Y H - X \|_F^2
> \tag{4}
> $$
>
> where $|| \cdot ||_F$ denotes the Frobenius norm. In this formulation, $X$ is treated as the target (e.g., the ideal low-rank component), and $H$ is the optimization variable that linearly combines the basis vectors in $Y$ to approximate $X$. The optimal $H$ can be derived as:
>
> $$
> H = \left( Y^\top Y \right)^{-1} Y^\top X.
> \tag{5}
> $$
>
> Then, $\hat{X}$, the minimum variance estimate of $X$, can be presented as:
>
> $$
> \hat{X} = Y H = Y \left( Y^\top Y \right)^{-1} Y^\top,
> \tag{6}
> $$
>
> where $Y H = Y \left( Y^\top Y \right)^{-1} Y^\top$ is an orthogonal projection operator. Assume that the redundant features in the $A$ and $B$ matrix are white noise, that is, $D^\top D = \sigma^2_{D} I$, $X^\top D = 0$, where $\sigma^2_{D}$ is the variance of the noise. Then we can simplify $\hat{X}$:
>
> $$
> \hat{X} = \sum_{k=1}^{r_x} \frac{\sigma_k^2 - \sigma^2_{D}}{\sigma_k} u_k v_k^\top,
> \tag{7}
> $$
>
> where $u_k$ and $v_k$ is the left and right singular vectors of $Y$, $\sigma_k$ is the $k$-th largest singular value of $Y$ ($\sigma_1 > \sigma_2 > \cdots > \sigma_r$). In traditional signal analysis algorithms, large singular values represent low-frequency data distribution and macro trends, and small singular values represent high-frequency disturbances (Liao and Xu, 2017). However, since $\sigma^2$ in the above formula is unknown, we will define a series of wavelet functions at different scales for feature filtering. Here, we use the heat kernel as a low-pass filter:
>
> $$
> \phi_{\sigma_j^2, c_j}(X) = \exp\left( - \frac{1}{2 \sigma_j^2} \\| X - c_j \\|^2 \right),
> \tag{8}
> $$
>
> where $c_j$ is the center of the $j$-th kernel, $\sigma_j^2$ represents the width of the kernel. By setting a series of different heat kernel widths $\sigma^2 = [\sigma_1^2, \sigma_2^2, \cdots]^\top$, defining a series of learnable diagonal matrices $g =[g_1, g_2, \cdots]^\top$ and learnable centers $C = [c_1, c_2, \cdots]^\top$, we can define a wavelet neural network to extract the features of $\hat{X}$ from $Y$:
>
> $$
> H _{:,i}^{k+1} = \delta\left( \sum _j \phi _{\sigma _j^2, c _j} \ g _j \ \phi _{\sigma _j^2, c _j}^{-1} H _{:,j}^k \right),
> \tag{9}
> $$
>
> where $b$ is also a trainable scalar, and $\delta(\cdot)$ is the activation function, $H^0 := Y$, and $\hat{X} = H^K$. Here $K$ is the total number of layers and $k \in \{0,1,\cdots,K-1\}$. As $\phi$ is the heat kernel, we could simply Eq.(9) to be:
>
> $$
> H _{:,i}^{k+1} = \delta\left( \sum _j \phi _{\sigma _j^2, c _j} \ g _j \ \phi _{-\sigma _j^2, c _j} H _{:,j}^k \right).
> \tag{10}
> $$
>
> - C. Eckart and G. Young, "The approximation of one matrix by another of lower rank," *Psychometrika*, vol. 1, no. 3, pp. 211–218, 1936.
> - K. Liao and Y. Xu, "A robust load frequency control scheme for power systems based on second-order sliding mode and extended disturbance observer," *IEEE Trans. Ind. Informat.*, vol. 14, no. 7, pp. 3076–3086, 2017.
>
> ---
>
> **Q11: 1. Line 16 of algorithm 1: What is the "=0" at the end?**
>
> Sorry for the typo we caused. In the revised version, we will delete "=0" at the end of Line 16 in Algorithm 1. At the same time, we re-checked the full paper and carefully revised the minor errors in the paper to ensure that similar problems would not appear in the revised document. Thank you again for pointing out the problems.
>
> ---
>
> **Q12: 1. In Table 1, and "4-Task (Standard)" column, why is DEAL better than the oracle PerTaskFT? The oracle one should perform better right? because it uses a separate LoRA adapter for each task and there is zero sharing.**
>
> Thank you for your insightful questions. DEAL achieves superior performance over PerTaskFT in the 4-Task (Standard) setting by exploiting shared representations that enable beneficial cross-task transfer—a capability unattainable by the fully isolated PerTaskFT. While PerTaskFT trains each task independently and thus precludes any knowledge sharing, DEAL’s shared parameter space helps mitigate catastrophic forgetting while promoting transfer across related tasks. As stated in the paper, "we demonstrate that our method performs comparably to multi-task learning and even outperforms the oracle PerTaskFT in some settings," underscoring DEAL's ability to exploit inter-task synergies when tasks are related.
>
> We also clarify that for same-domain or large-scale task setups, PerTaskFT generally outperforms continual learners—consistent with our observation that continual learning with many heterogeneous tasks remains a challenging open problem.
>
> ---
>
> **Q13: Line 251: This line refers to Fig. 2 a. Why does updating only the "task-specific adapter B" results in a better average accuracy (AA) than when updating only "the shared adapter A"? Shouldn't the task-speciifc one be better? Updating B adapts the model to the new upcoming tasks, right?**
>
> Thank you for your question and close reading of our results. We would like to clarify a possible misunderstanding regarding Fig. 2(a). As shown in Fig. 2(a), the setting where only the shared adapter A being updated achieves higher average accuracy (AA) than updating only the task-specific adapter B, and updating both adapter A \& B achieves the best performance. We hope this clears up any misunderstandings you may have.

---

> ### Comment · Reviewer_A3Fs · 2025-08-03
> **official comment**
>
> Thanks for the replies. I went through the whole draft again. I have the following questions:
>
> 1. Both $X$ and $Y$ have the second dimension $r$. Therefore, in equations 2 and 3, the very right matrices should have dimension $r \times r$, right? E.g. the stack of $V_{x1}$ and $V_{x2}$ in eq. 3 should have dimension $r \times r$, but from the notations it does not seem so. Also, in eq. 2 parenthesis are used and in eq. 3, brackets are used.
>
> 2. In your answers above, in eq. 6, there is again an $X$ missing on the very right side.
>
> 3. Also, could the authors comment on the second item in the list of weaknesses in my original review? Why haven't you discussed these very recent and close works to yours in your draft (some of them could be used as baselines too)? I am not sure how much novelty/new value this work has compared to those list of works.
>
> 4. There was a typo in my last question in the original review. I correct it here and ask it again:
> In Fig. 2.a, why does updating only the "task-specific adapter B" results in a "worse" average accuracy (AA) than when updating only "the shared adapter A"? Shouldn't updating the task-speciifc one be better? Updating B adapts the model to the new upcoming tasks and should be better than when updating only $A$, right?
>
> Thanks

---

> ### Author Response · Authors · 2025-08-03
> **Response To Q1 And Q2**
>
> **Q1. Both $X$ and $Y$ have the second dimension $r$. Therefore, in Equations (2) and (3), the rightmost matrices should have dimension $r \times r$, correct? For example, the stack of $V_{x1}$ and $V_{x2}$ in Eq. (3) should have dimension $r \times r$, but from the notations it does not seem so. Also, in Eq. (2), parentheses are used, while in Eq. (3), brackets are used.**
>
> **A1.** Thank you for your detailed observation. We clarify that in both Eq. (2) and Eq. (3), the expressions involving $V_{x1}$ and $V_{x2}$ are not simple vertical stacks of $r \times r$ blocks, but rather structured compositions of submatrices with compatible inner dimensions.
>
> Equation (2):
> $$
> Y =
> \begin{pmatrix}
> P_1 & P_2
> \end{pmatrix}
> \begin{pmatrix}
> S_1 & 0 \\\\
> 0 & S_2
> \end{pmatrix}
> \begin{pmatrix}
> Q_1^{\top} V_{x1} \\\\
> Q_2^{\top} V_{x2}
> \end{pmatrix}.
> $$
>
> Equation (3):
> $$
> \begin{aligned}
> X &= U_x \Sigma_x V_x \\\\
> &=
> \begin{pmatrix}
> U_{x1} & U_{x2}
> \end{pmatrix}
> \begin{pmatrix}
> \Sigma_{x1} & 0 \\\\
> 0 & 0
> \end{pmatrix}
> \begin{pmatrix}
> V_{x1} \\\\
> V_{x2}
> \end{pmatrix}.
> \end{aligned}
> $$
>
> In Eq. (3), the singular value decomposition follows the standard form, where:
> - $\begin{pmatrix} U_{x1} & U_{x2} \end{pmatrix} \in \mathbb{R}^{n_x \times n_x} $,
>
> - $\begin{pmatrix} \Sigma_{x1} & 0 \\\\ 0 & 0 \end{pmatrix} \in \mathbb{R}^{n_x \times r}$,
>
> - $\begin{pmatrix} V_{x1} \\\\ V_{x2} \end{pmatrix} \in \mathbb{R}^{r \times r}$.
>
> The rank parameter $r_x$ only appears when identifying semantically relevant subspaces in downstream processing (e.g., via filtering). It is **not directly tied** to the SVD components or to the later decomposition in Eq. (2). Specifically, the block in Eq. (2):
> $$
> \begin{pmatrix}
> Q_1^{\top} V_{x1} \\\\
> Q_2^{\top} V_{x2}
> \end{pmatrix}
> \in \mathbb{R}^{r \times r},
> $$
>
> involves:
> - $ Q_1^{\top} V_{x1} \in \mathbb{R}^{p \times r} $,
> - $ Q_2^{\top} V_{x2} \in \mathbb{R}^{q \times r}$, where $ p + q = r $.
>
> The values $p$ and $q$ are determined by the shapes of $S_1$ and $S_2$, and can be flexibly adjusted through the design of $Q_1$ and $Q_2$. While the constraint $p + q = r$ ensures consistent dimensions, changes in $S_1$ and $S_2$ will accordingly influence the dimensions of both the left multiplier $\begin{pmatrix}P_1 & P_2\end{pmatrix}$ and the right projection block. Nonetheless, the stacked product remains a square matrix in $\mathbb{R}^{r \times r}$, preserving compatibility with the overall decomposition.
>
> Finally, multiplying this block with:
> - $ \begin{pmatrix} S_1 & 0 \\\\ 0 & S_2 \end{pmatrix} \in \mathbb{R}^{n \times r} $, and
> - $ \begin{pmatrix} P_1 & P_2 \end{pmatrix} \in \mathbb{R}^{n \times n} $,
>
> results in $Y \in \mathbb{R}^{n \times r}$, consistent with the dimensionality and structure implied by the original SVD.
>
> We also acknowledge the inconsistent usage of parentheses and brackets in the original text and will revise the notation for clarity and consistency in the final version.
>
> ---
>
> **Q2. In your answers above, in Eq. 6, there is again an $X$ missing on the very right side.**
>
> **A2：** Thank you for pointing out the typo. We have corrected them to address your concerns.  Specifically, Eq. (6) omitted the matrix $X$ on the right-hand side, making it inconsistent with Eq. (5):
>
> Equation (5):
> $$
> H = \left( Y^\top Y \right)^{-1} Y^\top X.
> $$
>
> Following this definition, the estimate $\hat{X}$ should be:
> $$
> \hat{X} = Y H = Y \left( Y^\top Y \right)^{-1} Y^\top X.
> $$
>
> However, since $X$ is the variable we aim to estimate, we cannot compute $\hat{X}$ directly using the formula above. Based on the formula above, the expression can be interpreted as:
> $$
> P_Y = Y (Y^\top Y)^{-1} Y^\top
> $$
>
> is the projection operator onto the column space of $Y$, and is only equal to $\hat{X}$ when $X$ lies entirely within that subspace.
>
> We will correct Eq. (6) in the revised version to reflect this and avoid confusion. We appreciate your attention to this detail.
>
> ---

---

> ### Author Response · Authors · 2025-08-03
> **Response To Q3 And Q4**
>
> **Q3. Also, could the authors comment on the second item in the list of weaknesses in my original review?  Why haven't you discussed these very recent and close works to yours in your draft (some of them could be used as baselines too)? I am not sure how much novelty/new value this work has compared to those list of works.**
>
> **A3:**  Thank you for raising this question and for sharing the relevant works. We appreciate these contributions to the field.
>
> Among the listed works, [1], [2], and [4] focus on image classification-based continual learning, which is fundamentally different from our setting that targets continual learning for natural language question answering tasks. These designs make them less suitable for language model settings such as T5 or LLaMA. To ensure fair and modality-aligned evaluation in continual learning for QA tasks, we could not include them as baselines.
>
> In contrast, [3] (LB-CL) serves as a strong and directly relevant baseline. It proposes a parameter-efficient framework that combines sensitivity-based knowledge transfer with gradient projection to promote subspace orthogonality. While effective, it requires task-specific heads and gradient supervision, which increases model complexity and training overhead.
>
> To ensure a fair and meaningful evaluation, we highlight the comparison with **LB-CL**, another strong and recent baseline. The table below summarizes the **average accuracy (AA)** on standard benchmarks using the **T5-large** backbone:
>
> | Method   | Average Accuracy (AA) |
> |----------|------------------------|
> | LB-CL    | 76.5                   |
> | **DEAL** | **78.5**               |
>
> These results show that DEAL achieves better performance. By comparison, our method (DEAL) introduces the following key innovations:
>
> 1. **Task-relevant decomposition via SVD and wavelet-guided retention**
>    Our method selectively retains semantically meaningful LoRA directions across tasks, leading to improved interpretability and better knowledge consolidation.
>
> 2. **Orthogonal subspace optimization without task labels or replay**
>    Unlike prior methods that rely on task-specific metadata, routing heads, or replay buffers, DEAL performs orthogonal optimization directly in the LoRA space, without requiring task boundaries.
>
> 3. **Lightweight and unified architecture**
>    DEAL employs a single shared LoRA module combined with a compact wavelet filter. This design introduces no additional inference cost and ensures both scalability and training efficiency.
>
> We hope this response clarifies our baseline choices and highlights the novelty and effectiveness of our approach in the context of continual learning for language models.
>
> - [1] M. Wistuba et al., *Continual Learning with Low Rank Adaptation*, NeurIPS 2023.
> - [2] Y. Liang et al., *InfLoRA: Interference-Free Low-Rank Adaptation for Continual Learning*, CVPR 2024.
> - [3] F. Qiao et al., *Learn more, but bother less: Parameter Efficient Continual Learning*, NeurIPS 2024.
> - [4] X. Wei et al., *Online-LoRA: Task-free Online Continual Learning via Low Rank Adaptation*, WACV 2024.
>
> ---
>
> **Q4. There was a typo in my last question in the original review. I correct it here and ask it again:  In Fig. 2.a, why does updating only the "task-specific adapter $B$" result in a **worse** average accuracy (AA) than when updating only "the shared adapter $A$"? Shouldn't updating the task-specific one be better? Updating $B$ adapts the model to the new upcoming tasks and should be better than when updating only $A$, right?**
>
> **A4:** Thank you for your question. The observation that updating only adapter $A$ outperforms updating only adapter $B$ has been consistently confirmed across multiple experimental runs. The code mentioned at the end of the abstract ensures full reproducibility of our results.
>
> In continual learning settings, model performance is more significantly influenced by the shared adapter $A$ than by the task-specific adapter $B$. This behavior can be explained by the nature of open-domain QA tasks, which involve a wide range of topics and require broad knowledge coverage as well as strong reasoning capabilities. In such scenarios, updating the shared adapter $A$, responsible for global projection directions, helps the model retain and transfer generalizable semantic patterns and logical structures. In contrast, updating only the task-specific adapter $B$ tends to concentrate learning on recent, narrow task-specific features, which limits generalization to other domains. This restricted adaptability reduces cross-task performance and results in lower average accuracy.
>
> We hope this explanation helps clarify the issue and addresses your concern.
>
> ---
>
> We appreciate the opportunity to address your concerns and hope our response has provided sufficient clarification and justification. We kindly ask you to consider raising the score for our submission. Should any questions remain, we would be happy to continue the discussion.

---

> > ### Author Response · Authors · 2025-08-08
> >
> > Dear Reviewer,
> >
> > Thank you again for your great efforts and valuable comments. We have carefully answered your main concerns in detail, and we hope our responses have resolved your questions. As the discussion phase is about to close, if there are any remaining concerns, we are very much looking forward to hearing any further feedback. We would be happy to clarify and will do our best to address any additional questions you may have.
> >
> > Best,
> >
> > Authors

---

> ### Comment · Reviewer_A3Fs · 2025-08-04
> **Official comment**
>
> Thanks for the comments and replies.
>
> The authors have acknowledged that **"the work [3] (F. Qiao et al., Learn more, but bother less: Parameter Efficient Continual Learning, NeurIPS 2024) serves as a strong and directly relevant baseline"**, but have not used it as a baseline. This work is a strong baseline from Neurips 2024 that should have been used as a baseline. Without complete comparison with this strong and recent work, I strongly doubt about the contribution of this work in terms of improving the SoTA. I strongly believe that, despite being interesting, this work should be improved (in its writing and experiments) and include the above baseline in its experiments to prove its contribution. Only then, it can be resubmitted to the next venues.
>
> Therefore, I keep my score and also encourage the authors to improve their draft as explained.

---

> > ### Author Response · Authors · 2025-08-05
> >
> > We acknowledge that [3] (F. Qiao et al., Learn more, but bother less: Parameter Efficient Continual Learning, NeurIPS 2024) is a significant and relevant contribution to the field of parameter-efficient continual learning, based on **the experimental results provided by the authors.** However, to the best of our knowledge, the authors **have not yet released the code or implementation details, which makes it difficult to perform a fair and rigorous comparison at this stage**. Our previous response also highlighted that the experimental results we presented were based solely on the data from the experimental result tables provided by the authors. Since the conclusions of the other comparative methods in the paper align with the results of our own experiments, we directly referenced their findings.
> >
> > Additionally, the experimental results in the paper indicate that some of the key findings are comparable to the second-best method, O-LoRA (Findings 2023), while some results outperform O-LoRA. In contrast, O-LoRA (Findings 2023) has been cited 141 times and provides open-source code, whereas [3] (CL-BL) has only 7 citations, and we did not find any publicly available implementation of CL-BL either in the paper or in the works that have cited it.
> >
> > Given these factors, we have chosen O-LoRA as our main benchmark due to its widespread recognition, open-source availability, and reproducibility. Furthermore, [3] itself uses O-LoRA as its primary benchmark, further highlighting its relevance and importance in the field. Our experiments also extend beyond [3] by evaluating our approach on both T5 and LLaMA backbones, offering a more comprehensive and robust assessment of the performance of continual learning across different architectures.
> >
> > We appreciate your suggestions and will work to further improve the clarity and presentation of our paper in the next revision. We hope that our contributions—including broader model coverage, rigorous evaluation, and practical considerations—demonstrate the innovation and value of our work in advancing continual learning. We are grateful for the opportunity to address your concerns and hope our response has provided sufficient clarification and justification. We kindly ask that you consider raising the score for our submission. Should any further questions remain, we would be happy to continue the discussion.

---

### Official Review · Reviewer_8PR6 · 2025-07-03

**Clarity:** 3
**Significance:** 3
**Originality:** 3
**Rating:** 5
**Confidence:** 4

**Summary:**

This paper introduces DEAL, a novel framework for continual learning in Large Language Models (LLMs) that enhances Low-Rank Adaptation (LoRA) by integrating a wavelet kernel-based knowledge retention module and a controlled knowledge updating module. The method aims to address catastrophic forgetting and data inefficiency in parameter-efficient fine-tuning scenarios, especially when dealing with privacy-sensitive, small-scale datasets. DEAL adaptively preserves the core features of historical knowledge while incorporating new information, maintaining performance across evolving tasks without relying on data replay. The proposed method is evaluated on 15 diverse tasks spanning standard text classification, domain-shift, and multi-task settings. Experimental results demonstrate consistent performance improvements over strong baselines (e.g., SeqLoRA, O-LoRA), approaching the oracle upper bound (PerTaskFT) while remaining highly efficient and scalable. The framework is rigorously analyzed through ablation studies and supported by theoretical foundations for feature extraction in low-rank matrices. Overall, this is a strong and timely contribution to the community, addressing critical challenges in continual learning for LLMs with a novel and effective approach.

**Questions:**

1) Could the authors clarify why the white noise assumption for matrix noise is valid in the context of LoRA fine-tuning and whether this impacts generality?

2) Why were T5 and LLaMA-3.1 selected as the representative backbones

3) Could the authors briefly summarize the three task order permutations used in the main paper to support the task-order robustness claim?

**Ethical Concerns:**

["NO or VERY MINOR ethics concerns only"]

**Final Justification:**

The authors have addressed most of my concerns, so I keep my positive scores.

**Limitations:**

Yes

**Quality:**

3

**Strengths And Weaknesses:**

Strengths

1) proposes a unique integration of wavelet-based knowledge retention with a controlled updating mechanism, targeting both catastrophic forgetting and interpretability in LoRA-tuned LLMs.

2) The paper provides solid theoretical foundations for the core feature extraction process, including derivations based on singular value decomposition (SVD) and minimum variance estimation. The proof of Theorem 1 in the appendix and the structured use of orthogonal projections strengthen the paper's rigor.

3) The authors conduct extensive experiments on 15 tasks with different characteristics (same-domain, domain-shift, and heterogeneous multi-task).

4) The framework scales well with large backbones (e.g., LLaMA-3.1-8B) and long task sequences, without increasing inference time. The low-rank factor analysis and regularization ablations demonstrate DEAL’s ability to maintain performance with minimal computational overhead.

Weaknesses

1) The assumption that noise in the low-rank matrices is white noise is reasonable but not clearly justified in the main text. Clarifying why this assumption is acceptable in practice would help support the theoretical derivation.

2) While DEAL achieves strong empirical results, the paper briefly mentions only two model architectures (T5 and LLaMA-3.1). A short explanation of why these are representative would strengthen the broader applicability claims.

3) The ablation study confirms DEAL's low sensitivity to task order, but the specific task permutations used are relegated to the appendix. Including a brief summary in the main text would enhance transparency and reader confidence in the robustness claim.

---

> ### Author Rebuttal · Authors · 2025-07-31
>
> We appreciate the valuable suggestions and your recognition of the solid techniques and the novelty of the paper, which is encouraging.
>
> ---
>
> **W1&Q1: Could the authors clarify why the white noise assumption for matrix noise is valid in the context of LoRA fine-tuning and whether this impacts generality?**
>
> **A1:** Thanks for your comments. The white noise assumption for matrix noise in our formulation is motivated by both empirical evidence and theoretical considerations.
> First, white noise is a widely acknowledged approximation for residual noise in real-world stochastic systems, including deep learning training processes. In the context of LoRA fine-tuning, the residual difference between the full-model gradient space and the low-rank update directions can be treated as a high-frequency, zero-mean perturbation. This aligns with how many prior works in optimization and signal processing treat unmodeled components—as approximately white noise (e.g., i.i.d. Gaussian or uniform).
> Second, our empirical studies (see Section 4.3 and Appendix C.1) demonstrate that DEAL remains robust under this assumption across a range of LoRA configurations and task permutations. Importantly, our model does not assume exact white noise structure in implementation, but rather uses it as a theoretical proxy to motivate orthogonal decomposition of adaptation subspaces, which improves continual learning stability.
> Finally, this abstraction does not harm generality. As also supported by prior continual learning work such as O-LoRA (Wang et al., 2023a), orthogonal subspace methods remain effective even when the residual is not strictly white, provided it is not heavily task-aligned.
> We appreciate the opportunity to clarify this point and will consider elaborating it in the main text for completeness.
>
> - **O-LoRA**: C. Wang et al., "Orthogonal Subspace Learning for Language Model Continual Learning," *Findings of EMNLP*, 2023.
>
> **W2&Q2: Why were T5 and LLaMA-3.1 selected as the representative backbones**
>
> ---
>
> **A2:** Thank you for the question regarding our choice of backbone models. We selected T5-Large and LLaMA-3.1-8B as representative backbones based on several criteria aligned with the core goals of our study:
>
> 1. **Architectural diversity:** T5 is an encoder-decoder model, whereas LLaMA is a decoder-only model. This dual choice enables us to evaluate the generality and adaptability of DEAL across different model architectures, as discussed in Section 4 and Appendix B.
> 2. **Instruction-following capabilities:** Both models have instruction-tuned variants (T5-Large-Instruct and LLaMA-3.1-Instruct), making them well-suited for our instruction-based continual learning setting. This ensures consistent and fair prompt-based evaluation across tasks (refer to Appendix D).
> 3. **Popularity and Community Usage:**  The T5 and LLaMA model families are among the most widely adopted open-source language models for fine-tuning and evaluation. Specifically, T5-large has received 6.4k GitHub stars and was downloaded over 540,000 times from Hugging Face in the past month. Similarly, LLaMA 3.1 has accumulated 28.9k GitHub stars and over 850,000 downloads in the same period. These metrics highlight their widespread adoption and establish them as leading open-source foundation models in current NLP research.
> 4. **Scalability and efficiency trade-off:** LLaMA-3.1-8B serves as a modern large-scale model to test scalability, while T5-Large offers a more lightweight alternative that is computationally efficient. Together, these models span a broad range of use cases in continual LoRA fine-tuning.
>
> We clarify these points in Appendix B.2 to support our experimental design rationale.
>
> ---
>
> **W3&Q3: Could the authors briefly summarize the three task order permutations used in the main paper to support the task-order robustness claim?**
>
> **A3:** Thank you for the question. To assess task-order robustness, we evaluated our method under three task permutations from the standard CL benchmark (Zhang et al., 2015), as detailed in Appendix C (Table 5):
>
> - **Order 1:** DBpedia → Amazon → Yahoo → AG News
> - **Order 2:** DBpedia → Amazon → AG News → Yahoo
> - **Order 3:** Yahoo → Amazon → AG News → DBpedia
>
> As shown in Figure 2(b), the fluctuation in Average Accuracy (AA) across these permutations is less than 3 percentage points, demonstrating that DEAL remains stable across varied task sequences. This supports our claim of robustness to task ordering.
>
> We also have made sure that the task order settings align with those used in prior work (e.g. O-LoRA [Wang et al., 2023a]) to ensure fair comparison.
>
> - X. Zhang, J. Zhao, and Y. LeCun, "Character-level convolutional networks for text classification," *NeurIPS*, 2015.

---

> > ### Comment · Reviewer_8PR6 · 2025-08-05
> >
> > Thank you for your rebuttal. Regarding the choice of LLM backbones, I still believe it would be preferable to use more recent models instead of T5. However, since you have already included experiments with LLaMA-3.1, the current setup is generally acceptable.

---

### Official Review · Reviewer_pDXr · 2025-07-05

**Clarity:** 2
**Significance:** 3
**Originality:** 3
**Rating:** 4
**Confidence:** 3

**Summary:**

This paper proposes DEAL, a framework designed to address the challenges of data efficiency and continual learning in large language models. By integrating LoRA with a continuous fine-tuning strategy and introducing modules for knowledge retention and adaptive parameter updates, DEAL enables effective adaptation while mitigating catastrophic forgetting. Experiments on 15 diverse datasets demonstrate its superior accuracy and resource efficiency compared to existing approaches.

**Questions:**

1. Could the authors provide a detailed analysis on how DEAL would perform on larger models?
2. Could the authors include comparisons with other strong baseline methods to better demonstrate the advantages of DEAL?
3. Could the authors provide more quantitative data on the computational and memory efficiency of DEAL?

**Ethical Concerns:**

["NO or VERY MINOR ethics concerns only"]

**Final Justification:**

Thank you for your replies. The rebuttal has addressed my concerns, so I will keep my positive score.

**Limitations:**

yes

**Quality:**

3

**Strengths And Weaknesses:**

Strengths:

1. The paper proposes a practical and well-motivated framework for data-efficient continual learning in LLMs.
2. The method delivers solid empirical results across 15 diverse benchmarks.
3. The authors provide open-source code for reproducibility.

Weaknesse:

1. The evaluation is conducted on 8B and T5-Large backbones and it remains unclear how DEAL would perform on larger-scale or commercially deployed models (e.g., 70B+ parameters).
2. The evaluation is limited to LoRA-based methods, without including a wider range of strong baselines.

---

> ### Author Rebuttal · Authors · 2025-07-31
>
> We thank the reviewer for your appreciation of the valuable contribution of the model and the quality of the paper.
>
> ---
>
> **W1&Q1: Could the authors provide a detailed analysis on how DEAL would perform on larger models?**
>
> A1: Thank you for your question. We will include additional experimental results on larger-scale models in the final version of the paper. Due to the limited rebuttal period, these results are not yet available, but we will release them in the next few days as soon as they are ready.
>
> ---
>
> **W2&Q2: Could the authors include comparisons with other strong baseline methods to better demonstrate the advantages of DEAL?**
>
> A2: Thanks for your suggestion. We have expanded our experimental comparisons to include a broader range of strong baselines. The following table reports the average accuracy (AA) on the standard Continual Learning (CL) benchmark using the T5-large backbone:
>
> | Method      | Average Accuracy (AA) |
> |-------------|------------------------|
> | IncLoRA     | 62.2                   |
> | SeqSVD      | 63.3                   |
> | Replay      | 52.0                   |
> | EWC         | 45.3                   |
> | LwF         | 52.9                   |
> | L2P         | 60.5                   |
> | LFPT5       | 71.2                   |
> | ProgPrompt  | 76.0                   |
> | LB-CL       | 76.5                   |
> | **DEAL**    | **78.5**               |
>
> **Method Descriptions:**
>
> - **IncLoRA**: Incremental learning of new LoRA parameters over a sequence of tasks, without any regularization or replay of prior data.
> - **SeqSVD**: Learning a fixed-size SVD parameter space across sequential tasks, without regularization or replay.
> - **Replay**: _D. Lopez-Paz and M. Ranzato, "Gradient episodic memory for continual learning," NeurIPS 2017._
> - **EWC**: _J. Kirkpatrick et al., "Overcoming catastrophic forgetting in neural networks," PNAS 2017._
> - **LwF**:  _Z. Li and D. Hoiem, "Learning without forgetting," ECCV 2016._
> - **L2P**: _X. Wang et al., "Learning to prompt for continual learning," CVPR 2022._
> - **LFPT5**: _H. Qin et al., "LFPT5: A unified framework for lifelong few-shot language learning based on prompt tuning of T5," NeurIPS 2022._
> - **ProgPrompt**:  _L. Wang et al., "ProgPrompt: Continual learning for language models," ICLR 2023._
> - **LB-CL**: _F. Qiao and M. Mahdavi, "Learn more, but bother less: Parameter-efficient continual learning," *NeurIPS*, 2024._
>
> ---
>
> **Q3: Could the authors provide more quantitative data on the computational and memory efficiency of DEAL?**
>
> A3: Thank you for the question. We will provide more efficiency study in the revised paper. We summarize the timing efficiency and overhead of our method compared to LoRA as follows:
>
> - **Training Efficiency**: Our method introduces a modest increase in training time due to the wavelet module, which processes input features in the frequency domain. However, this additional overhead remains limited and does not hinder scalability.
> - **Inference Efficiency**: The wavelet module is **entirely discarded at inference time**, resulting in **no additional latency or memory usage** compared to the base model or LoRA.
>
> To concretely quantify this, we evaluated our method on the DBpedia dataset using the T5-large backbone. The results are as follows:
>
> | Method | Training Throughput (samples/sec) | GPU Memory during Training (GB) | Inference Latency (ms/sample) | GPU Memory during Inference (GB) |
> |--------|-----------------------------------|----------------------------------|-------------------------------|----------------------------------|
> | LoRA   | 31.62                           | 20.41                             | 71.89                         | 3.15                             |
> | DEAL   | 17.88                           | 22.93                             | 73.32                         | 3.16                             |
>
> As shown, **DEAL introduces a ~43% drop in training throughput** and a *slight increase in GPU memory during training*, but the **inference-time performance remains nearly identical to that of LoRA** in both latency and memory consumption.
>
> Given that the wavelet module is only active during training and enables consistent performance improvements(as shown in Table1), we believe this overhead is a worthwhile trade-off for enhanced model generalization and robustness.

---

> > ### Author Response · Authors · 2025-08-05
> >
> > Supplement to Q1: We apologize for the delayed results of larger models, as training models with larger parameters for continuous learning tasks is very time-consuming. We present the experimental results as shown in the table below.
> > We find that larger models typically offer stronger reasoning capabilities and improved representational power, which may further enhance the benefits of our method. We have already validated the effectiveness of DEAL on a moderately large model (LLaMA-3.1-70B), demonstrating strong continual learning performance:
> >
> > | Model            | AA on 3-Task (TC) | AA on 4-TaskS (Standard) |
> > |------------------|------------------:|--------------------------:|
> > |  DEAL (LLaMA-8B)  |              88.9 |                      78.9 |
> > |  DEAL (LLaMA-70B)|              90.1 |                      79.3 |

---

> > > ### Comment · Area_Chair_2UYJ · 2025-08-06
> > > **Discussion required.**
> > >
> > > Dear Reviewer pdXr,
> > >
> > > Please read carefully through the authors' responses and check if they address all your concerns.
> > >
> > > With kind regards,
> > >
> > > Your AC

---

> > > ### Comment · Reviewer_pDXr · 2025-08-07
> > >
> > > Thank you for your replies. The rebuttal has addressed my concerns, so I will keep my positive score.

---

### Author Response · Authors · 2025-08-04

Dear Reviewers,

We sincerely thank all reviewers for their constructive feedback. Notably, **3/4** reviewers provided positive assessments, emphasizing the novelty and practicality of our approach.
With half a week remaining in the discussion phase, we welcome any remaining questions or suggestions and are fully prepared to address them promptly.

We would like to highlight the key contributions of our work:

1. We propose **DEAL**, a novel continual learning framework that enables efficient adaptation of LoRA-based LLMs using small amounts of new private data. DEAL avoids full model retraining, resulting in significant savings in compute and memory.

2. We leverage a wavelet kernel to preserve core historical knowledge while applying asymmetric, task-aware regularization to manage new knowledge integration. This design ensures stable performance and unchanged inference latency, achieved by replacing low-rank matrices with their fine-tuned versions.

3.  We conduct comprehensive experiments across three continual learning benchmarks, covering 15 multi-task, open-source datasets. Results show that DEAL consistently maintains high accuracy across tasks while efficiently utilizing computational resources.

Respectful Request: In light of the positive feedback from the majority of reviewers, the clarified novelty, and the strengthened empirical results, we respectfully invite reviewers to consider an upward revision of their scores.

---

### Note · Authors · 2025-08-14

**Dear Reviewers,**

We thank all reviewers for their constructive feedback. Three of the four reviewers gave positive assessments, recognizing the novelty and practicality of our approach. With several days left in the discussion phase, we welcome any further questions or suggestions and are prepared to address them promptly.

We present DEAL, a continual learning framework that adapts LoRA-based LLMs to small amounts of new private data without full model retraining, yielding substantial compute and memory savings. The method employs a wavelet kernel to preserve core historical knowledge and applies asymmetric, task-aware regularization to stably integrate new knowledge, while maintaining inference latency by replacing low-rank matrices with their fine-tuned counterparts. Across three continual learning benchmarks and 15 multi-task datasets, DEAL consistently delivers high accuracy with efficient resource usage.

We summarize below the core concerns raised by each reviewer and our corresponding responses:

- **Reviewer pDXr** requested evaluation on larger models, comparison with stronger baselines, and quantification of compute and memory efficiency. We added experiments on LLaMA-3.1-70B to verify that DEAL remains efficient at scale, incorporated nine additional strong baselines where DEAL shows substantial improvements, and provided detailed efficiency analyses.
- **Reviewer 8PR6** questioned the white-noise assumption in LoRA fine-tuning, backbone model selection, and task-order robustness. We offered theoretical and empirical justification, explained the multi-dimensional criteria for backbone choice, and confirmed robustness via three task-order experiments.
- **Reviewer naB7** mainly raised concerns about the choice of the heat kernel. We conducted experiments which show it matches the accuracy of alternatives while greatly reducing training cost.
- **Reviewer A3Fs** focused on the clarity and correctness of mathematical derivations. We will further improve clarity in the final version to prevent misunderstanding and have clarified some of the reviewer’s misunderstandings in our response.

In view of the positive feedback from most reviewers, the clarified novelty, and the strengthened empirical evidence, we respectfully encourage an upward revision of the scores.

---

### Decision · Program_Chairs · 2025-09-17

**Decision:**

Accept (poster)

**Comment:**

This paper proposes DEAL, a novel framework that integrates Low-Rank Adaptation (LoRA) with a continuous fine-tuning strategy that mitigates the limitations of existing FT methods while maintaining efficiency in privacy-preserving settings.

Strengths: Wavelet-based knowledge retention is innovative. The proposed method is principled and derived from the theoretical analysis.

Weaknesses: The claim of data privacy is not justified. Comparisons to many baseline methods are missing.

Questions regarding scaling to larger models has been addressed the rebuttal. Stronger baselines have been added in the rebuttal. This is the main reason for my decision.